# Mitochondrial origins of fractional control in regulated cell death

Luís C. Santos[1], Robert Vogel[2,3], Jerry E. Chipuk[1,4], Marc R. Birtwistle [5,6], Gustavo Stolovitzky [2,3] & Pablo Meyer [2,3]

Individual cells in clonal populations often respond differently to environmental changes; for binary phenotypes, such as cell death, this can be measured as a fractional response. These types of responses have been attributed to cell-intrinsic stochastic processes and variable abundances of biochemical constituents, such as proteins, but the influence of organelles is still under investigation. We use the response to TNF-related apoptosis inducing ligand (TRAIL) and a new statistical framework for determining parameter influence on cell-to-cell variability through the inference of variance explained, DEPICTIVE, to demonstrate that variable mitochondria abundance correlates with cell survival and determines the fractional cell death response. By quantitative data analysis and modeling we attribute this effect to variable effective concentrations at the mitochondria surface of the pro-apoptotic proteins Bax/Bak. Further, our study suggests that inhibitors of anti-apoptotic Bcl-2 family proteins, used in cancer treatment, may increase the diversity of cellular responses, enhancing resistance to treatment.

[1] Department of Oncological Sciences, Icahn School of Medicine at Mount Sinai, New York, NY 10029, USA. [2] IBM T.J. Watson Research Center, 1101 Kitchawan Road, Yorktown Heights, NY 10598, USA. [3] Department of Genetics and Genomic Sciences, Icahn School of Medicine at Mount Sinai, New York, NY 10029, USA. [4] The Tisch Cancer Institute, Icahn School of Medicine at Mount Sinai, 1 Gustave L. Levy Place, New York, NY 10029, USA. [5] Systems Biology Center New York, Icahn School of Medicine at Mount Sinai, New York, NY 10029, USA. [6] Department of Chemical and Biomolecular Engineering, Clemson University, Clemson, SC 29634, USA. These authors contributed equally: Luís C. Santos, Robert Vogel. Correspondence and requests for materials should be addressed to M.R.B. (email: mbirtwi@clemson.edu) or to G.S. (email: gustavo@us.ibm.com) or to P.M. (email: pmeyerr@us.ibm.com)

sogenic populations of cells in homogeneous environments have the seemingly paradoxical capacity to generate many unique cell states. This ability is found in many, if not all, types of single-celled organisms and in the distinct cell types of multicellular organisms. For example, *Bacillus subtilis* cells were shown to independently and transiently switch between vegetative and competent states[1], hematopoietic progenitor cells can differentiate into either erythroid or myeloid lineages[2], and cancerous tissue maintain distinct subpopulations throughout the course of disease[3]. A cell's propensity for a particular state is attributed to the intrinsic stochasticity of low-copy number biomolecular reactions[4–6] or extrinsic variations in the abundances of its components, in all such cases[7–9]. Taken together, it is clear that stochastic transitions of cell state, that are driven by non-genetic sources of cell-to-cell variability (CCV), are fundamental to the maintenance of single-cell populations, the function of distinct tissues, and structure of clinical lesions in diseases such as cancer.

One commonly studied source of CCV is protein abundance. Its premier status as a dominant source of non-genetic CCV is due to its stochastic production[6,10], and the sensitivity of cellular decision-making machinery to variations in their components. For example, in biological signal transduction, information regarding the cell's environment is processed by a cascade of biomolecular reactions. Variation from one cell to another in any one of the corresponding biomolecules varies the signal magnitude across the population, making unique the cell's perception of environmental conditions and its corresponding response[11–14]. While it has been definitively shown that CCV in protein abundance influences cellular decisions, little attention has been given to other non-genetic sources of CCV.

There are numerous examples in which non-genetic and non-protein sources of CCV are conjectured to impact biological phenomena. For example, centrosome abundance[15], the size of the Golgi apparatus[16], and mitochondria abundance[17–20] all have been shown to vary from cell to cell. To determine if diversity in cell behaviors may be attributed to CCV in organelle abundance, our study focuses on the role of mitochondria in the context of TNF-related apoptosis-inducing ligand (TRAIL)-induced apoptosis.

Indeed, the abundance of mitochondria per cell has been shown to positively correlate with a cell's propensity for apoptosis[20]. The mechanism of this phenomena was attributed to CCV in protein abundances, which were previously shown to correlate with mitochondria abundance[21]. However, in this study we show through the analysis and interpretation of TRAIL dose response curves that, in fully TRAIL-responsive cell lines, cell survival correlates with a higher density of mitochondria and a large portion of the CCV in cell death can be attributed to variations in mitochondria abundances influencing the effective concentrations of Bax/Bak on the mitochondrial surface.

## Results

**Mitochondria density correlates with resistance to TRAIL.** To assess whether mitochondria abundance correlated with single-cell sensitivity to TRAIL-induced apoptosis (Fig. 1a), we measured the binary life-or-death status and the abundance of mitochondria of individual cells by flow cytometry. During extrinsic apoptosis, TRAIL stimulates cell death by binding to its cognate death receptors on the cell surface, forming a complex that activates Caspase 8 (Fig. 1a), the so-called initiator caspase (IC). Active IC activates pro-apoptotic BH3-only proteins, which, directly or indirectly, activate pro-apoptotic Bcl-2 family proteins Bax/Bak. Active Bax/Bak can commit a cell to apoptosis by translocating from the cytosol to the outer mitochondrial membrane, where they oligomerize and form pores[22,23], which allow for the diffusion of pro-apoptotic molecules from the intermembrane space of the mitochondria into the cytosol[24,25]. The pro-apoptotic activities of Bax/Bak are counteracted by pro-survival Bcl-2 proteins such as Bcl-xL, which constantly retro-translocates Bax/Bak from the mitochondria back into the cytosol[26,27], thus protecting cells from committing to apoptosis by shifting relative subcellular localization of Bax/Bak[28,29]. In effect, these molecules dynamically regulate each other's activity so that the continuous values of TRAIL concentration can be converted to a binary dead-or-alive response.

The human T-lymphoblastoid leukemia-derived cells (Jurkat), a human breast adenocarcinoma cell line (MDA-MB-231), and HeLa cells were exposed to different doses of TRAIL for 4 h, a time frame in which cells died readily but the single-cell mitochondria abundances remained largely unchanged (Supplementary Notes 1 and 2, Supplementary Figs. 1–4). For each dose of TRAIL, we measured in single cells the abundance of mitochondria and the cell state by concomitant labeling with a fluorescent Annexin V and MitoTracker Deep Red and analysis of flow cytometry measurements (FCM). Living cells, Annexin V negative and MitoTracker high are well separated from the dead cells, Annexin V positive and MitoTracker low and medium (Fig. 1b, see Supplementary Fig. 1 for complete gating strategy). Importantly, the fact that the living and apoptotic cell populations shared almost no MitoTracker population led us to conclude that the apoptosis process corrupted the MitoTracker signal. Consequently, the apoptosis process precludes assessment of mitochondria abundance by MitoTracker in Annexin V-positive cells.

From the FCM and our live cell gate we confirmed that Jurkat and MDA-MB-231 cell lines were sensitive to TRAIL (Fig. 1c, f), but HeLa cells were not as responsive (see Supplementary Fig. 11). Furthermore, from fitting the Hill model to each dose response we found that these cell lines had vastly different sensitivities ($IC_{50}$) to TRAIL, $3.81 \pm 0.26$ ng/mL for Jurkat cells, $76.4 \pm 8.77$ ng/mL for MDA-MB-231 cells and more than 300 ng/mL for HeLa cells. To facilitate comparison between cell lines, we color coded the effective TRAIL dose so that we may track the mitochondria abundance with the effective, as opposed to the experimental, dose of TRAIL (Fig. 1c, h and Supplementary Fig. 11).

Next, we found that mitochondria abundance of living cells is correlated with cell size, as measured by forward scatter (FSC) (Fig. 1d, g). To eliminate analyzing effects due to cell size, as opposed to mitochondria, we focused our attention to the mitochondria density, $\rho$, defined as the MitoTracker signal normalized to FSC signal. With these data we estimated the probability density of single-cell mitochondria density in live cells for each dose of TRAIL. Here we found that with successively increasing doses of TRAIL the probability distribution of $\rho$ becomes increasingly enriched for cells with high mitochondria density (Fig. 1e, h). Moreover, we found that the degree of the enrichment is unique to each cell line—Jurkat cells were more readily biased in their mitochondria density than were MDA-MB-231 and HeLa cells (for all HeLa cell analysis see Supplementary Note 5 and Supplementary Fig. 11).

We hypothesize that the observed enrichment of cells with high mitochondria density is established by a differential sensitivity of single cells to TRAIL. An intuitive result considering that the sensitivity of a signaling pathway to its cognate ligand is tuned by the abundances of its components. In apoptosis for example, we would expect that the number of TRAIL receptors on a cell's surface, the number of pro-caspase molecules, the number of Bax/Bak molecules, the number of mitochondria, etc. contribute to that cell's response to a single dose of TRAIL. If each one of these molecules varied from one cell to the next, the so-called

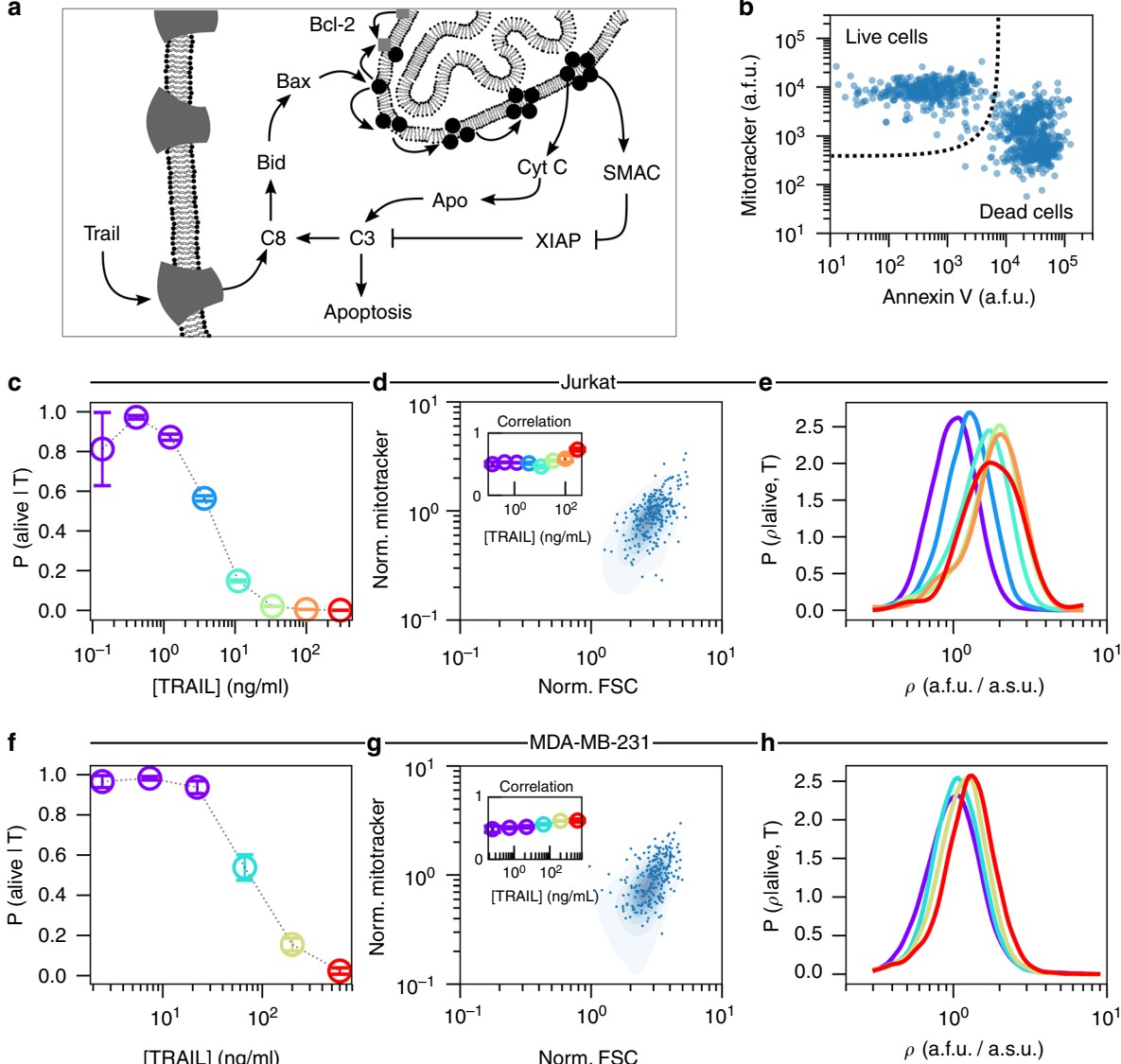

**Fig. 1** TRAIL administration enriches for cells with high density of mitochondria. **a** An overview of TRAIL-induced apoptosis. **b** Flow cytometry measurements (FCM) of mitochondria (MitoTracker Deep Red) and phosphatidylserine (FITC-conjugated Annexin V) in Jurkat cells. Complete flow cytometry gating strategy can be seen in Supplementary Fig. 1. The fractional response of Jurkat cells (**c**) to TRAIL. Each color corresponds to a unique fractional response to a specific TRAIL dose. Cell size measurements (FSC-A) in Jurkat cells (**d**) are correlated with mitochondria abundance (MitoTracker Deep Red). The inset shows that the Pearson correlation marginally changes for each TRAIL dose. The probability density of mitochondria density ($\rho$) for each dose of TRAIL that elicits a unique response in Jurkat cells (**e**). The fractional response of MDA-MB-231 cells to TRAIL (**f**). Cell size measurements (FSC-A) in MDA-MB-231 cell (**g**) are correlated with mitochondria abundance (MitoTracker Deep Red). The inset shows that the Pearson correlation marginally changes for each TRAIL dose. The probability density of mitochondria density ($\rho$) for each dose of TRAIL that elicits a unique response in MDA-MB-231 cells (**h**). In (**e**) and (**h**) the single-cell measurements from each of the lowest three doses of TRAIL are aggregated prior to probability density estimation (Violet). Visual inspection of the respective dose response curves suggests that these three doses of TRAIL are effectively identical. Data presented with error bars represent the mean ± one standard error of the mean over triplicate experiments

CCV, we should expect that the individual response of cells to TRAIL are unique.

Indeed, the probability density of $\rho$ shows that the endogenous density of mitochondria varies from cell to cell (Fig. 1e, h). If each cell's sensitivity to TRAIL were anticorrelated with mitochondria abundance, we would expect an enrichment of high mitochondria density cells with TRAIL stimulation. Such an effect can be quantitatively studied by using probability considerations. By applying Bayes' theorem we may associate the changes in the probability density $\rho$ with the quantitative change in the fraction of living cells. From this simple property of probability, we were able to develop a quantitative strategy to gauge whether the observed endogenous variability of biological components are responsible for functional population diversity.

**Variability in all-or-none biological responses**. As found in other biological systems, e.g. MAPK and NFκB[30,31], the conversion of a continuous input to a binary response limits the influence of CCV in cellular components to CCV in sensitivity to perturbations. Also, previous methods have been developed to determine how non-genetic CCV of protein levels influences the fractional responses of cell fate decision in cell populations during mitotic checkpoint signaling[32] and apoptosis[33], but none takes

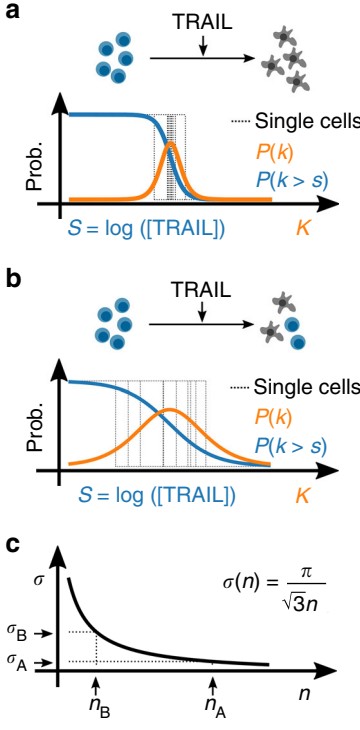

**Fig. 2** Cell-to-cell variability in the binary response to TRAIL. Hill response function with respect to TRAIL dose (blue) and the corresponding probability density of the single-cell sensitivities (orange) for populations with **a** low CCV and **b** high CCV. **c** The theoretical correspondence between the variance of single-cell sensitivities to TRAIL ($\sigma$) and the Hill coefficient $n$. Here, ($n_A$, $\sigma_A$) and ($n_B$, $\sigma_B$) represent the Hill coefficient and corresponding single-cell variances from **a** and **b**, respectively

advantage of the information available in a full dose response curve to perturbations. In apoptosis, each cell, with its unique concentrations of molecular components, should require a specific concentration of TRAIL to induce cell death. At the population level the diversity in single-cell sensitivities to TRAIL gives rise to the fractional control of cell death.

As an example, consider two separate ensembles of cells, one with near-identical biomolecular composition (low CCV) and the other with variable numbers of its components (high CCV). In the scenario where all components are near equal, the individual cells will undergo the life–death transition at nearly the same dose of ligand (Fig. 2a). In contrast, when CCV is relatively high, the individual cells of the ensemble will transition from live to dead at diverse doses of TRAIL (Fig. 2b). The resulting fractional control of the population response to TRAIL would then take a steep or gradual sigmoid shape, respectively.

This interpretation of the empirical dose response curve represents the cumulative distribution of single-cell sensitivities, from which we may derive the corresponding probability density of single-cell sensitivities. Indeed, from this simple interpretation, the empirical dose response curve of binary biological responses contains a complete statistical description of the functional diversity in the population. Fitting this dose response to a Hill function, we find that the mean sensitivity of single cells to a perturbation is simply the logarithm of the $IC_{50}$, and the variance of single-cell sensitivities to be inversely proportional to the squared Hill coefficient (Fig. 2c).

Matching the dose response parameters to statistical quantities is useful because now we may use the tools of probability theory to analyze our data such as taking conditional moments. In the context of TRAIL-induced apoptosis, we may ask what is the

average sensitivity of cells given a specific mitochondria density. This statistical question is equivalent then to asking how does the $IC_{50}$ of individual cells changes with the mitochondria density. Or we may ask, what is the variance of single-cell sensitivities given that we measured mitochondria density, which, intuitively, quantitatively measures the remaining diversity in the population once we remove the contribution of mitochondria density. With this information we may then compute the fraction of the functionally relevant population diversity attributable to a measured component.

**Decomposing sources of CCV.** Let us assume that the sensitivity of cells to TRAIL is wholly dependent on the biological components of the apoptotic signaling pathway. For simplicity let us designate the mitochondria density $\rho$ as $x_0$ and all other contributing components as $x_1, x_2, \ldots, x_m$. A priori any mathematical function that describes the intricate relationships of these components and the dose of TRAIL to the single-cell sensitivity ($\kappa$) is unknown; however, we may expand this a priori unknown function to an arbitrary order by

$$\kappa = \log(IC_{50}) + \sum_{i=0}^{m} k_i \delta \log(x_i) + \cdots \quad (1)$$

where $k_i = \partial \kappa / \partial \log(x_i)|_{\langle x_i \rangle}$ and $\delta \log(x_i) = \log(x_i) - \log(\langle x_i \rangle)$ (see Supplementary Note 3 for details). The order in which we expand to will dictate the degree of complexity we wish to understand. If we limit our understanding to first order, then the details of the specific pathway are bundled into phenomenological parameters $k_i$. If then we infer $k_i$ from data, we can estimate the extent to which each component contributes to a cell's sensitivity to TRAIL.

Indeed, Eq. (1) provides a framework for constructing a single-cell interpretation of the Hill model, which incorporates the abundance of biological components with the stimulation strength. The biological species are introduced into the Hill model parameters by their influence on the first and second statistical moments of single-cell sensitivities, $\kappa$. For example, incorporating our measurements of mitochondria density $\rho$ to the $IC_{50}$ amounts to computing the average sensitivity conditioned on mitochondria density,

$$\log(IC_{50}(\rho)) = \log(IC_{50}) + k_\rho \delta \log(\rho). \quad (2)$$

Then, in like fashion, the resulting Hill coefficient comes from estimating the variance of sensitivities conditioned on mitochondria density,

$$n_\rho = \frac{\pi}{\sqrt{3 \sum_{i=1}^{m} \sigma_{\kappa:i}^2}} \quad \text{with} \quad \sigma_{\kappa:i}^2 = k_i^2 \sigma_i^2 \quad (3)$$

where $\sigma_i^2$ represents the variance of the $i$th biological component and $\sigma_{\kappa:i}^2$ is the variance of $\kappa$ attributed to species $i$. Note that in Eq. (3) the sum is from 1 to $m$ and consequently the diversity in single-cell sensitivities attributable to mitochondria are absent. In consequence, the removal of sources of CCV manifests as a smaller term in the denominator of Eq. (3) and consequently larger Hill coefficient. If we apply the moments from our first-order expansion Eqs. (2) and (3) to the Hill model, we arrive at our single-cell Hill model,

$$P(\text{alive}|\rho, T) = \frac{(\rho / \langle \rho \rangle)^{k_\rho n_\rho}}{(\rho / \langle \rho \rangle)^{k_\rho n_\rho} + ([T] / IC_{50})^{n_\rho}} \quad (4)$$

Equation (4) gives us a detailed understanding of the influence of mitochondria density, or in general any measured components. If, for example, mitochondria density does not contribute to the

cell's sensitivity to TRAIL then $k_\rho = 0$ and Eq. (4) reduces to the standard dose response Hill function. If however $k_\rho$ is not zero and positive, then mitochondria density effectively promotes cell survival, and if $k_\rho$ is negative then it increases the effectiveness of TRAIL. Together we can probe the influence of each measured component at unprecedented resolution. We call this strategy DEPICTIVE, which is an acronym for DEtermining Parameter Influence on Cell-to-cell variability Through the Inference of Variance Explained.

To see this in detail, let us consider an example of an arbitrary pathway consisting of five components that takes $s$ as input and provides a binary output $y$ (Fig. 3a). For each dose $s$ of the synthetic stimulus we generate virtual single-cell FCM (Fig. 3b) consisting of a subset of each cell's constituents ($x$, $z$, and $q$). Using these data, we compute the population response to the stimulus, and from the single-cell nature of the data interrogate the influence of each molecular constituent. In Fig. 3c we observe the influence of each constituent on the response. First, we see that the distribution of biological species $x$ does not change when we subset single cells upon their state $y$ (Fig. 3c). Consequently, the contribution of species $x$ will negligibly contribute to each cell's sensitivity to $s$, a fact corroborated by $P(y = 1|x, s)$ being weakly dependent on the abundance of $x$. Unlike species $x$, species $z$ and $q$ do influence each cell's behavior, which is apparent in analyzing the single-cell data. Intuitively, the changes of the distribution of molecular components conditioned by the cell state are the signal required for inferring each parameter $k_i$ from Eq. 1.

The inference each of the $k_i$ in the simulation data is trivial, because we have measurements of each biological component for each cell state $y$. Uniquely, our experimental data consist of MitoTracker measurements from live cells exclusively. This was because MitoTracker Deep Red signal is dependent on the electro-chemical properties of the mitochondria, which are different for live and dead cells. To infer the values of $k$ from such data we developed a new inference strategy for semi-supervised logistic regression and embed it as a module within the DEPICTIVE statistical framework (see Supplementary Note 3.3 for details). We apply our method to the synthetic data that is analogous to our measurements, that is the measurements associated with each virtual cell with a binary label $y = 1$. Quantitatively, we see that we can infer the constants $k_i$ (Fig. 3d), the corresponding variances explained (Fig. 3e), and lastly the dependence of the single-cell sensitivities on each biological component (Fig. 3f).

**Mitochondria density is a source of CCV**. We apply our new statistical framework, DEPICTIVE, to quantitatively dissect the dependence of single-cell sensitivities to TRAIL with mitochondria density (Fig. 1e, h). We see that the fractional response of the Jurkat cells to each dose of TRAIL, $P(\text{alive}|\rho, T)$ is strongly dependent on mitochondria density (Fig. 4a, see Supplementary Figs. 5–7 for goodness-of-fit analysis). Moreover, we see that the single-cell dose response curve translates from low TRAIL to high TRAIL doses with increasing mitochondria density (Fig. 4b). The MDA-MB-231 cells' fractional response (Fig. 4c, d) is less steep than that of Jurkat, indicating that the single-cell sensitivities of MDA-MB-231 cells to TRAIL are not as sensitive to CCV in mitochondria density as Jurkat (see Supplementary Figs. 8–10 for

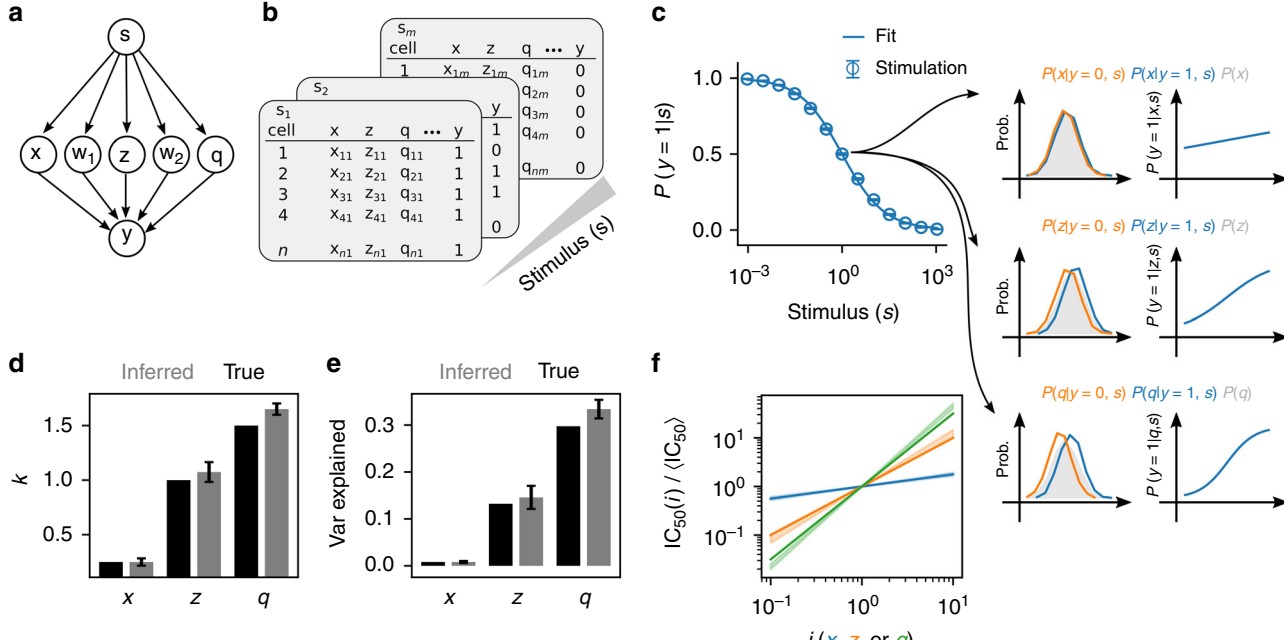

**Fig. 3** Decomposing sources of cell-to-cell variability. **a** Schema for a simple cellular response, $y \in \{0, 1\}$, to the activation of pathway components $w_1$, $w_2$, $x$, $z$, and $q$ representing the natural logarithm of abundances of each biological constituent subject to a dose of stimulus $s$. **b** Single-cell data were simulated to demonstrate the feasibility of CCV decomposition by sampling virtual cells (see Supplementary Note 3.3.4 for details). **c** The dose response of $N = 10,000$ simulated cells and $n = 100$ replicate experiments in which $k_x = 0.25$, $k_{w_1} = 0.5$, $k_z = 1$, $k_q = 1.5$, and $k_{w_2} = 2$. Error bars represent ± one standard deviation about the mean. The probability density of the components given the cell state being live or dead from a single replicate experiment reveals how the logarithm of abundances of each biological entity correlates with cell state (left column). We further examine the dependence of cell survival on TRAIL by examining the probability of the cell state, $y = 1$, given the dose, the DEPICTIVE inferred parameters associated with a single replicate simulation, and the abundance of each biological component (right column). Quantitative assessment of the true (black) and the average DEPICTIVE inferred (gray) parameters $k_x$, $k_z$, $k_q$ ± one standard deviation (**d**) and the corresponding variance explained by each component (**e**). **f** The scaling of the $IC_{50}(i)$ with $i = x$, $z$, or $q$ ± one standard deviation

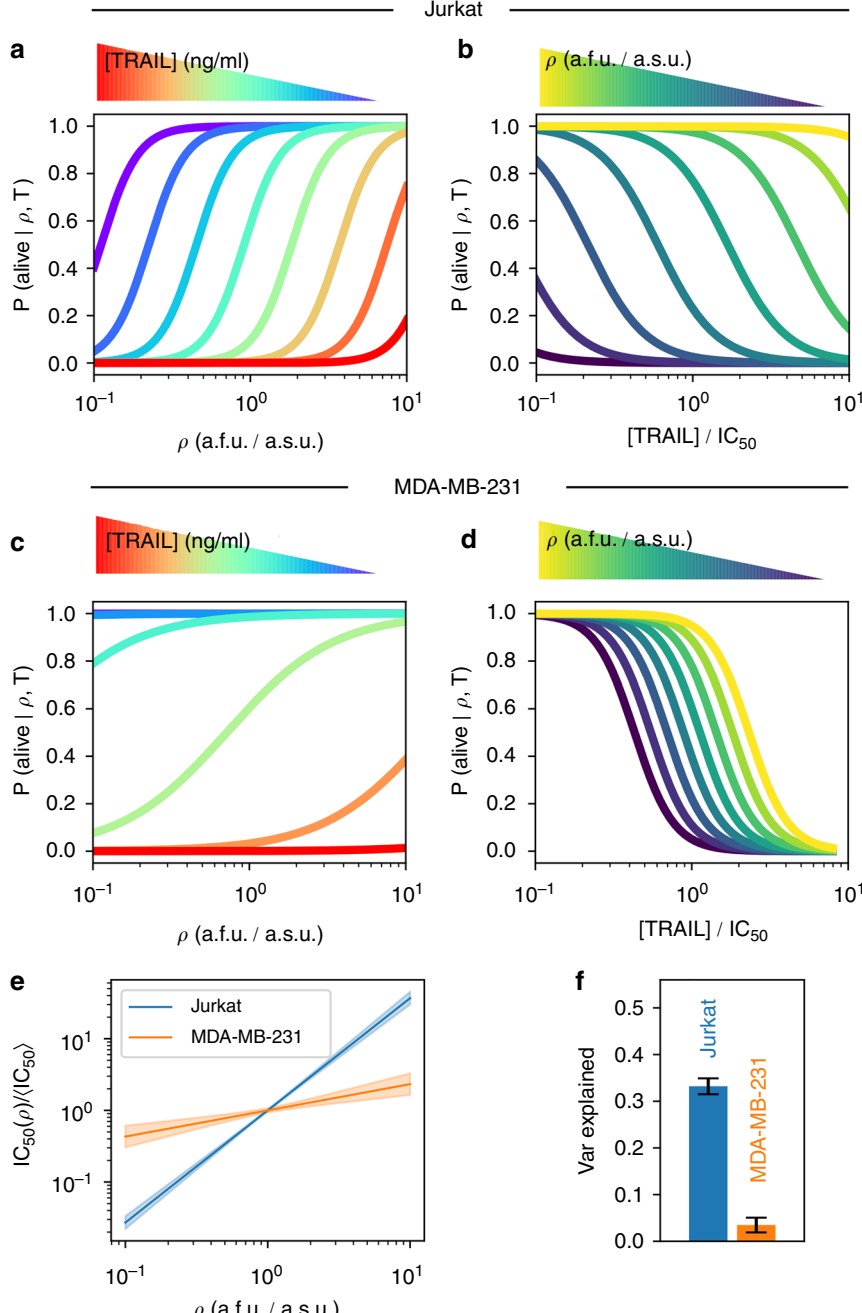

**Fig. 4** CCV in mitochondria density influences fractional response to TRAIL. The inferred fractional response determined by the single-cell Hill model and the average model parameters, over triplicate experiments, for Jurkat cells (**a**, **b**) and MDA-MB-231 cells (**c**, **d**) as a function of $\rho$ and TRAIL dose, respectively. (**e**) The dependence of single-cell sensitivities to TRAIL as a function of mitochondria abundance computed from the inferred model parameters. **f** The fraction of the variance in single-cell TRAIL sensitivities explained by CCV in mitochondria density. Error bars represent standard error of the mean of experimental triplicates. Detailed analysis of each replicate set are presented in Supplementary Figs. 5–10

goodness-of-fit analysis), a result that can be summarized by plotting the $IC_{50}(\rho)$ for each cell line (Fig. 4e). Moreover, we find that 30% and 2% of the diversity in single-cell sensitivities to TRAIL may be attributed to mitochondria density in Jurkat and MDA-MB-231 cells, respectively, while less than 2% is attributed to side scatter (SSC) that functions as an internal control (Fig. 4f and Supplementary Note 4 Supplementary Tables 1–4). Importantly, surviving cells with higher mitochondria density have stable levels of Bak at all TRAIL doses, indicating that mitochondria density is indeed the important factor for cell survival (Supplementary Figs. 14 and 15).

**Bax/Bak concentration depends on mitochondria surface area**. To gain mechanistic insight into the functional role of mitochondria density in the cell death decision, we developed a coarse-grained dynamic model of apoptosis (Fig. 5a). Our description aims to reproduce the dominant dynamical features of initiator caspase reporter protein (IC-RP) first measured and published by ref. [24], these being a slow but accelerating initial increase of IC followed by a fast increase in both IC and the Effector Caspase (EC). To such end our model includes (i) a slow auto-catalytic increase in IC activation, (ii) a quasi-steady-state approximation for Bax/Bak pore formation dynamics and

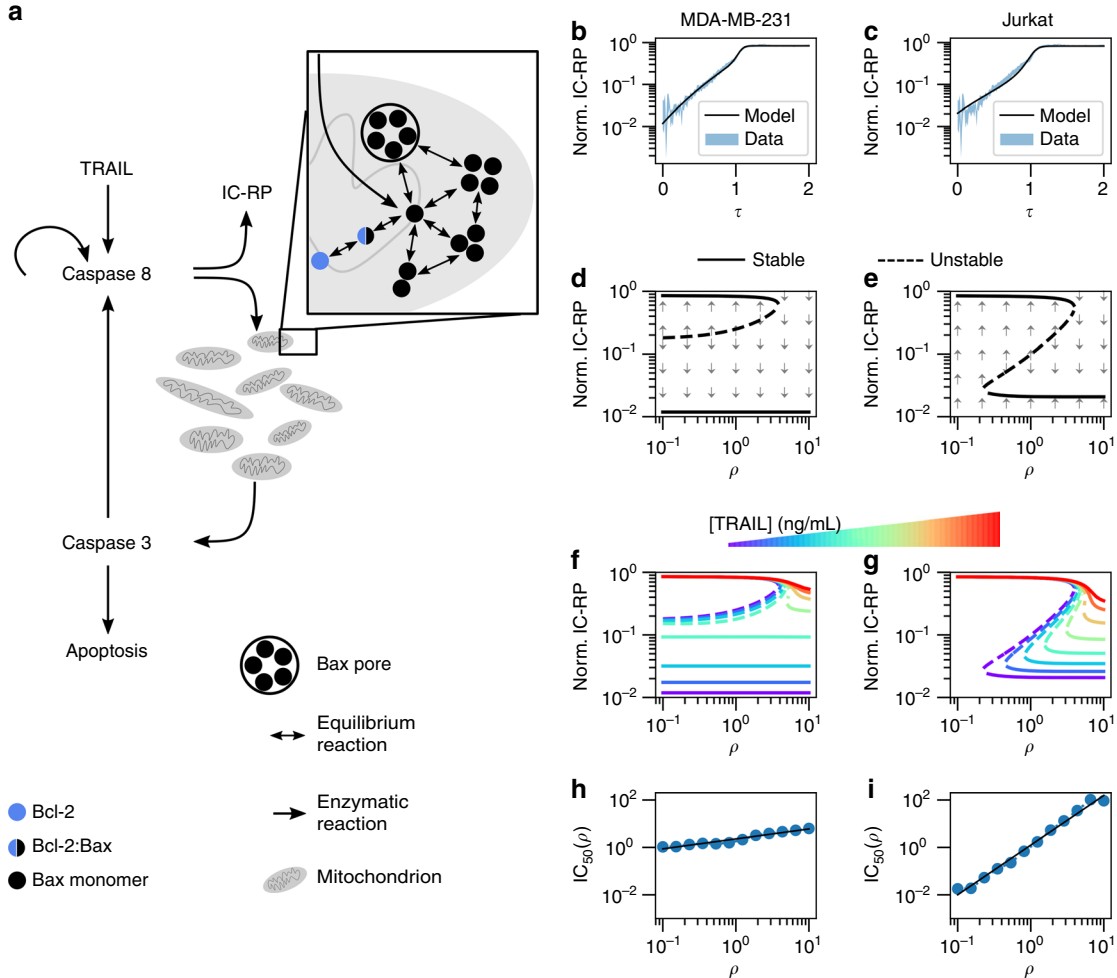

**Fig. 5** Mechanism of IC$_{50}$ dependence on mitochondria density. **a** Simple model of apoptosis. The dynamics of initiator caspase reporter protein (IC-RP) from ref. [24] and the model-inferred dynamics corresponding to **b** MDA-MB-231 and **c** Jurkat cell lines. The model bifurcation diagrams for [TRAIL] = 0 ng/ mL in **d** MDA-MB-231 and **e** Jurkat cells. The influence of TRAIL dose on the model fixed points for **f** MDA-MB-231 and **g** Jurkat cells. The dependence of single-cell sensitivities to TRAIL on $\rho$ for **h** MDA-MB-231 and **i** Jurkat cells. The IC$_{50}$ was estimated from Hill function fits of simulated data (blue circles), which were then fit to a power law (black line). Simulations consisted of 100 cells per 20 doses of TRAIL and 12 densities of mitochondria considered

mitochondrial outer membrane permeabilization (MOMP), and (iii) the strong positive feedback from EC to IC (see Supplementary Note 6 for details, Supplementary Tables 6 and 7 for variable definitions and Supplementary Table 8 and Supplementary Figs. 12 and 13 respectively for model parameter values and inference approach).

We conjectured that TRAIL-induced activation of IC in Jurkat and MDA-MB-231 cells match the biphasic increase of IC-RP measured in HeLa cells[25], but both differ in their propensity to form Bax pores (Fig. 5b, c). Specifically, we consider the unique susceptibilities of Bax/Bak pore formation to Bcl-2-mediated inhibition for each cell line. As Bax/Bak pores reside in the mitochondria, the effective Bax/Bak concentration for a given amount of Bax/Bak decreases with mitochondria density. Implementing this insight into the model equations we see that the influence of mitochondria density can be understood through the corresponding bifurcation diagrams (Fig. 5d, e).

The dynamic properties of IC in MDA-MB-231 cells in the absence of TRAIL are either bistable or monostable depending on mitochondria density. In these diagrams, the high IC fixed point corresponds to cells that have integrated sufficient signal for MOMP and consequently represent apoptotic cells. Cells with relatively low mitochondria density are bistable and may undergo apoptosis only if their IC abundance exceeds a critical amount

designated by the dashed line (Fig. 5d). This bistable region does not preclude cell death—cells may acquire sufficient abundances of IC for death by fluctuations in biomolecular reactions. Indeed, the likelihood of such an event decreases with the difference of IC abundance between the unstable fixed point (dashed line) and low IC stable fixed point (solid line). Meanwhile, cells with relatively high mitochondria density only have a single fixed point of low IC, indicating that these cells will never spontaneously undergo apoptosis in the absence of TRAIL.

In contrast, the bifurcation diagram representing Jurkat cells shows three distinct regions (Fig. 5e): (1) cells with low density of mitochondria having a single fixed point of high IC, where consequently all die; (2) cells with medium density of mitochondria that are bistable, for which the fractional response to TRAIL decreases with the concomitant increase in the IC unstable fixed point and mitochondria density; and (3) cells with high density of mitochondria that are monostable with low IC abundances, and hence all cells survive. Next, we extend these analyses to the full range of TRAIL doses.

The influence of increasing TRAIL dose in each cell type-specific parameterized model is evident in their bifurcation diagrams. MDA-MB-231 cells respond to TRAIL by increasing the IC abundance of the lower fixed point (Fig. 5f). In doing so, cells with mitochondria density in the bistable region equally

increase their susceptibility to cell death from fluctuations in IC abundance. The Jurkat model's response to TRAIL exhibits an increase of the density of mitochondria that separates the monostable high and bistable IC abundance regions (Fig. 5g). Therefore, an individual cell's mitochondria density determines its sensitivity to TRAIL-induced cell death. Together, these model-based observations propose an explanation for how CCV in mitochondria density influences the response of Jurkat but to a lesser extent MDA-MB-231 cells to TRAIL (Fig. 5h, i).

**Sensitizing MDA-MB-231 cells to CCV in mitochondria density.** While inspecting the model parameters associated with each cell type, we noticed that MDA-MB-231 cells were more susceptible to Bcl-2-mediated inhibition of Bax/Bak pore formation than Jurkat. We hypothesized that this effect would be abated by incorporating a small-molecule inhibitor to Bcl-2 in MDA-MB-231 cells (Fig. 6a, see Supplementary Note 6.1 for derivation). By incorporating Bcl-2 inhibition, we found that the sensitivity of the fractional response of the cell population to TRAIL increases (Fig. 6b). Furthermore, and as intuited, Bcl-2 inhibition increased the dependence of single-cell sensitivities to TRAIL on mitochondria density (Fig. 6d). We corroborated these theoretical predictions by measuring the influence of the clinically relevant small-molecule inhibitor of Bcl-2 family proteins ABT-263[34] (Fig. 6c, e, see Supplementary Figs. 8–10 for goodness-of-fit analysis). Remarkably, Bcl-2 inhibition alone increased the variance of sensitivities attributable to mitochondria density from about 2% to between 10 and 25% (Fig. 6f), but had no effect on HeLa cells (see Supplementary Table 5) although for the lower values the variance attributed to the internal SSC control was

barely lower (see Supplementary Note 4.2 and Supplementary Tables 2–4).

## Discussion

We have unveiled a connection in the CCV of mitochondria density to the fractional control of TRAIL-induced cell death. Importantly, we find that the dependency of single-cell sensitivities to CCV in mitochondria abundance is cell line dependent. Presumably this dependence originates in the unique composition of components across cell lines. In that, the functional manifestation of CCV in mitochondria on the sensitivity of single cells to TRAIL-induced apoptosis is dependent on the relative abundance and diversity of mitochondria in relation to the other biological constituents in the apoptosis pathway (see Supplementary Fig. 16 for BH3-like protein levels in different cell lines). Indeed, Jurkat cells readily responded to TRAIL and its $IC_{50}$ scaled with mitochondria abundance, MDA-MB-231 cells showed scaling and responded readily to TRAIL only in the presence of a pan-Bcl-2 inhibitor, while scaling was never observed and only a minority of HeLa cells responded to TRAIL even during Bcl-2 inhibition. Consequently, the seemingly contradictory results of our study and thoset of Márquez-Jurado[20] are manifestations of the unique biological systems being studied. In particular, it is possible that Márquez-Jurado's observations could be highly modified when considering mitochondrial density and not mass given that it is known that gene expression levels can change with cell size. We also think that our observations are highlighting a different phenomenon than Márquez-Jurado et al., who are measuring a mitochondrial mass dependence in HeLa cells that have died as a function of TRAIL dose, but show that, as in our setup, the mean

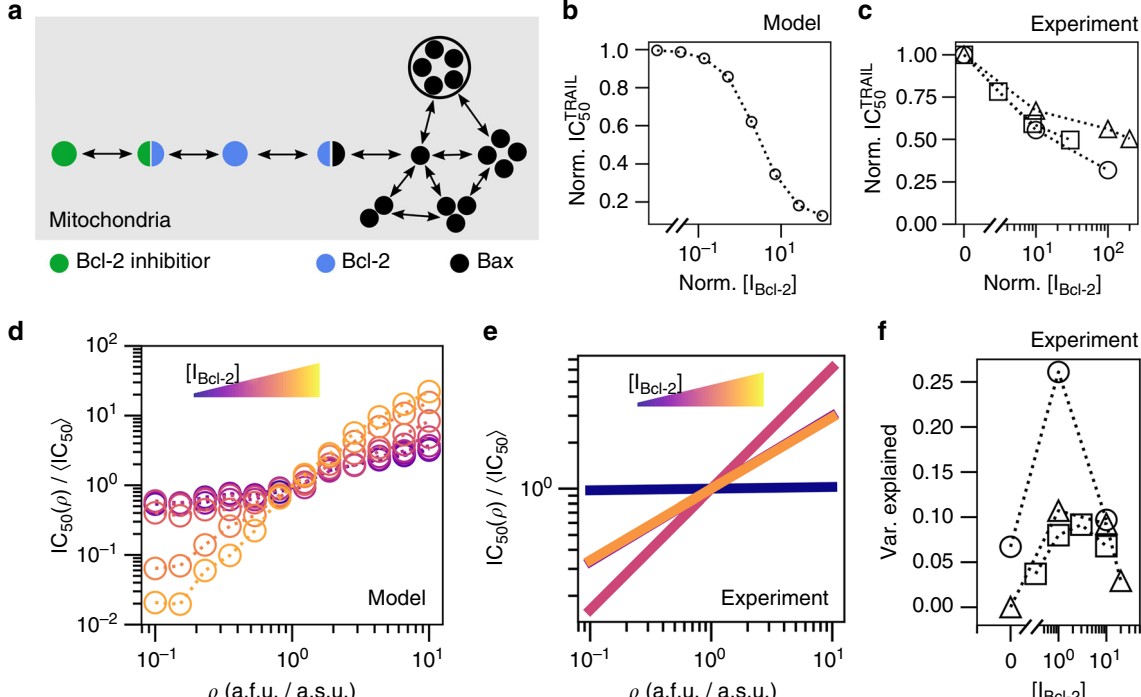

**Fig. 6** Plasticity in fractional response to TRAIL. **a** Bcl-2 inhibitor reduces the effective abundance of Bcl-2 by formation of Bcl-2:Bcl-2 inhibitor complex. **b** Simulation results of the population $IC_{50}^{TRAIL}$ response to Bcl-2 inhibition in MDA-MB-231 cells. **c** Experimental measurement sets—uniquely represented by a square, circle or triangle marker—of the population$IC_{50}^{TRAIL}$ response to Bcl-2 inhibition for MDA-MB-231 cells. **d** Estimated $IC_{50}$ for changing $\rho$ from MDA-MB-231 parameterized model simulations. **e** The experimental dependence of $IC_{50}$ on $\rho$, from a single representative experiment of three replicate experiments (Supplementary Figs. 8–10), as computed in Fig. 4e for [0, 1, 10, 20] μM doses of the Bcl-2 Inhibitor ABT-263. **f** The fraction of the observed diversity in single-cell sensitivities explained by mitochondria density CCV from triplicate experiments. Note that all simulations were conducted with 100 cells for each of the 20 doses of TRAIL, 12 densities of mitochondria, and 9 doses of inhibitor

mitochondria abundance among the live cells remains relatively constant and independent of the dose (see Fig. 2a in their manuscript and Supplementary Note 5 and Supplementary Fig. 11). In addition, we infer that mitochondria density being the main driver for cell survival is also complementary to results showing that Bax/Bak mitochondrial localization determines apoptosis[28,29,35].

Our findings were established by a new statistical framework, DEtermining Parameter Influence on Cell-to-cell variability Through the Inference of Variance Explained, namely DEPICTIVE, we developed to measure the impact of CCV on the binary response of cells to perturbation. It is composed of two parts, the first part is to infer the parameters of the logistic regression model when data from one or both of the binary cell state labels are available, while the second part provides the mathematical bases for interpreting the logistic regression model parameters to compute useful quantities.

Indeed, inferring the parameters of a logistic regression model from data is commonplace. However, it is only commonplace when data representative of both of the corresponding binary states are well established. To our knowledge, there is no method to infer these parameters from data where only one of the binary classes is readily available. In our study, data from live and dead cells were unavailable because our experimental label of mitochondria abundance, MitoTracker Deep Red, was not reliable for dead cells.

The second part of DEPICTIVE statistical framework is to use the logistic model parameters to estimate the contribution of the measured biological component(s) to the variable binary response of single cells. Applying this tool, we found that mitochondria density accounts for nearly 30% of the variable response to TRAIL in Jurkat cells and varies from 2% to up to 25% in MDA-MB-231 cells when Bcl-2 is inhibited. Conversely, HeLa cells showed no mitochondrial density dependence. Together, the two parts of the DEPICTIVE statistical framework can extract quantitative insights into sources of CCV.

We attribute the measured connection of TRAIL sensitivity and mitochondria density to the dilution of Bax/Bak on the outer mitochondrial membrane in cells by mathematical modeling. From the quantitative insights of DEPICTIVE, we found that the functional manifestation of mitochondrial CCV is plastic—readily and predictably tunable by small-molecule inhibitors of Bcl-2. It is plausible that this plasticity is a tool accessible to cells, and therefore may be co-opted by pathological cellular populations. For example, high mitochondria abundance can be a non-genetic mechanism of resistance to pro-apoptotic therapeutics. Incorporation of such knowledge may be an important consideration in developing therapeutic strategies, such as combination therapies.

The observed advantage of cells with high mitochondria densities may manifest in time scales much longer than the life span of a single cell or the disease in a human, but propagate to the long time scales of evolution. To date, the evolutionary hypothesis of mitochondria is as a symbiotic bacterium inside a proto-eukaryotic cell[36], exchanging safety for energy. However, another such evolutionary advantage may be expected, that this symbiosis would create a survival advantage such as the one described here. These results suggest that environmental constraints can select subpopulations not only based on genetic composition, protein abundances, but also based on CCV in organelle abundances.

## Methods

**Cell culture**. Jurkat E6-1 cells originate from a male human acute T-cell Leukemia and were purchased from ATCC (TIP-152). Cells were cultured in RPMI-1640 medium (Corning cat. 10-040-CV) supplemented with 10% heat-inactivated fetal bovine serum (Corning cat. 35-011-CV), 2 mM L-glutamine (Corning cat. 25-005-CI), and 1 mM sodium pyruvate (Corning cat. 25-000-CI). Cells were cultured at 37 °C in 5% $CO_2$ in a humidified incubator and maintained at a cell density not exceeding $3 \times 10^6$ by addition of fresh medium, or by centrifugation with subsequent resuspension at $1 \times 10^5$ cells/mL.

MDA-MB-231 cells originate from a human female adenocarcinoma that was harvested from a metastatic site in the breast and purchased from ATCC (no. HTB-26). Cells were cultured in DMEM medium (Corning cat. 10-017-CV) supplemented with 10% fetal bovine serum and 2mM L-glutamine (Corning cat. 25-005-CI). Cell were cultured at 37 °C in 5% $CO_2$ in a humidified incubator and subcultured every 2–3 days with 0.25% trypsin (Corning cat. 25-053-CI) to maintain sub-confluent density.

HeLa cells were purchased from ATCC (ATCC CCL2). Cells were cultured in DMEM medium (Corning cat. 10-017-CV) supplemented with 10% fetal bovine serum and 2 mM L-glutamine (Corning cat. 25-005-CI). Cells were cultured at 37 °C in 5% $CO_2$ in a humidified incubator and subcultured every 2–3 days with 0.25% trypsin (Corning cat. 25-053-CI) to maintain sub-confluent density.

**Apoptosis assay and data acquisition**. Jurkat cells were pelleted by centrifugation for 5 min at $100 \times g$, and then resuspended in 1× PBS and stained with 200 nM MitoTracker Deep Red (Life Technologies, cat. M22426) for 10 min at 37 °C. MitoTracker staining was quenched with full cell culture medium, followed by centrifugation for 5 min at $100 \times g$. Cells were resuspended in cell culture media at a density of $1 \times 10^6$ per mL, in which $1 \times 10^5$ were transferred to each experimental well of a flat-bottom 96-well plate. Cells were then incubated at 37 °C for 4 h with different doses of Superkiller TRAIL (Enzo Life Sciences cat. ALX-201-115) and/or ABT263 (ApexBio cat. A3007). After drug treatment, cells were transferred to a v-bottom 96-well plate, pelleted by centrifugation at $1000 \times g$, stained with FITC-conjugated Annexin V (Biolegend cat. 640945), and then measured by flow cytometry.

MDA-MB-231 or HeLa cells were seeded on 12-well plates at $5 \times 10^5$ cells per well in 400 μL, incubated overnight at 37 °C in 5% $CO_2$ in a humidified incubator until 80% confluent. Cells were then washed once with PBS and stained with 200 nM MitoTracker Deep Red (Life Technologies, cat. M22426) for 10 min at 37 °C. MitoTracker staining was quenched with full cell culture medium, and then incubated at 37 °C for 4 h with different doses of Superkiller TRAIL (Enzo Life Sciences cat. ALX-201-115) and/or ABT263 (ApexBio cat. A3007). After drug treatment, the supernatant containing floating cells was collected, and the remaining adherent cells were trypsinized, pooled with the supernatant, and pelleted by centrifugation for 5 min at $1000 \times g$. Cells were then stained with FITC-conjugated Annexin V (Biolegend cat. 640945), and then measured by flow cytometry.

FCM were conducted on a BD LSRII maintained by the Icahn School of Medicine at Mount Sinai flow cytometry core facility.

**FCM gating**. FCM were gated as follows: to exclude debris (Supplementary Fig. 1A), then gated for singlets (Supplementary Fig. 1B), MitoTracker Deep Red positive (Supplementary Fig. 1C), and lastly for living cells by Annexin V (Supplementary Fig. 1D). The fraction of cells alive was computed by dividing the number of cells in the Annexin-V-negative gate by the number of cells of the MitoTracker Deep Red-positive gate. Subsequent single-cell analysis was then conducted exclusively using cells from the Annexin-V-negative gate.

**Modeling and statistical analysis**. Detailed derivations of our DEPICTIVE statistical framework, application of DEPICTIVE to data, dynamic models, and inference of dynamic model parameters can be found in Supplementary Notes 3–6 and 6.4, respectively.

**Reporting summary**. Further information on experimental design is available in the Nature Research Reporting Summary linked to this article.

## Code availability

DEPICTIVE: Detailed derivation of the DEPICTIVE strategy can be found in Supplementary Note 3. We developed a user-friendly Python package to run the DEPICTIVE analysis strategy. The code is freely available as a GitHub repository, https://github.com/DEPICTIVE. Along with these tools we provide two tutorials that demonstrate how to generate synthetic data and to apply DEPICTIVE analysis. Dynamics simulations: Detailed derivations and parameter values of model equations for simulation can be found in Supplementary Note 6. We developed a user-friendly Python package to runour model. The code is freely available as a GitHub repository, https://github.com/DEPICTIVE. Along with these tools we provide a series of tutorials that demonstrate the use of our tools by examples. These tutorials can be found on the repositories' wiki pages, https://github.com/DEPICTIVE.

## Data availability

The data presented in the main text of this paper can be found on Mendeley data[37–43].

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

## Acknowledgments

This research was funded in part by grants from the NIH, (P50GM071558 (Systems Biology Center New York), R01GM104184 and U54HG008098 (LINCS Center) for M.R.B., R01 CA206005 for J.E.C., and P30 CA196521, a Tisch Cancer Institute Cancer Center Support Grant) and also an IBM faculty award for M.R.B. We wish to thank John Albeck for sharing single-cell data and staff of the flow cytometry core facility at ISMMS, Ravi Iyengar for useful discussions, and Julia Moore Vogel for help with the manuscript.

## Author contributions

Conceptualization, L.C.S., R.V., M.R.B., G.S. and P.M.; methodology, R.V., G.S. and P.M.; software, R.V. and P.M.; formal analysis, R.V. and G.S.; investigation, L.C.S., R.V. and P.M.; resources, J.E.C., M.R.B. and G.S.; writing—original Draft, L.C.S, R.V. and P.M.; writing—review & editing, L.C.S., R.V., M.R.B, G.S. and P.M.; visualization, R.V.; supervision, M.R.B., G.S. and P.M.

## Additional information

**Competing interests:** The authors declare no competing interests.

