## [Peer Review File · Nature Communications]

Reviewers' comments:

Reviewer #1 (Remarks to the Author):

In the manuscript "Origins of fractional control in regulated cell death", the authors describe an elegant statistical model relating cell-to-cell variability to observed survival curves. The proposed method allows the authors to assess the importance of individual measured factors, e.g. protein abundances, on the observed variability in the binary phenotype. The approach is employed to study the response to TRAIL in two cell lines.

The problem of relating molecular and phenotypic data is very important. I congratulate the authors to a great contribution to the field. The method, DEPICTIVE, is novel and appealing, amongst others, due to its simplicity. The authors present a simulation study as well as a study using real experimental data that indicate the robustness of the approach. As fractional data are very common in the testing of drug responses, I think that the proposed analysis approach will become very valuable. In addition to a statistical assessment, the authors show that it can provide mechanistic insights.

Overall, the manuscript is well written and mostly easy to follow. I have only a few minor questions and suggestions that should be very easy to address:

1) There are some studies describing fractional data using mechanistic models, e.g. Heinrich et al., Nat. Cell Biol., 15(10), 2013 and Kallenberger et al., Sci. Signal., 7(316), 2014. I think it would be good to discuss this work. It appears that the existing methods are also rather flexible but more complex and do not possess the same beauty and easy applicability.

2) The assumption that the mitochondria abundance is more or less constant seems to be rather important for the application. For this reason, I would suggest to improve Supplement Figure 2. In B and D, that the high population seems to drop below the control and it is unclear how the variability changes. To convince readers that the above assumption is correct, I would suggest to depict the full distributions at the individual time points after sorting. (In addition, the lines shown in D might be removed. The parameterization is unclear and I have a hard time imagining that the best fit is shown.

3) The experimental data seem to provide merely information about the CCV in mitochondria density. Accordingly, the possibility of DEPICTIVE to select / rank the most important sources cannot be exploited. It would have been interesting to see this. One simple possibility would be to include forward and side scatter in the experimental data, two parameters which likely to be uninformative.

4) In the simulation test shown in Figure 3, x_i seems to be normally distributed. However, in the derivation it was stated that in experiments the distribution is more log-normal. It would be interesting to run the test in this setting. In this context, I was also wondering why the authors did not expand κ on page 6 in the Supplement directly with respect to $\log(x)$.

5) Supp. p.6: Are the subscript in the equation before Eq. 9 correct or should "m" be " ρ "? For the derivation of (17) some additional comments might be valuable.

6) Supp. p.7: I think it would be helpful to explain why the IC20 values were chosen.

7) Supp. Fig. 3 and later: Labeling seems to be inconsistent with text. " $\pi()$ " is used instead of " $p()$ ".

8) The mechanistic model presented in the section "Bax concentration dependence on mitochondria surface area" possesses several parameters and it is not really clear how well the

parameters can be estimated from the available data.

In the context of reusability I was wondering whether the authors plan to release some software package. This would definitely boost the impact further and would allow other researchers to continue along this profitable route

I waive my right to anonymity,
Jan Hasenauer

Reviewer #3 (Remarks to the Author):

Meyer and colleagues study cell-to-cell variability to TRAIL-induced apoptosis in a statistical approach. Seemingly overwhelmed by the large body of Bax literature the authors missed a number of essential publications altogether. Previously, variable effective concentrations of the pro-apoptotic protein Bax at the mitochondria have been shown to determine the fractional cell death response (Todt et al., 2013 and Todt et al., 2015) and to correlate with cell survival of leukemia patients (Reichenbach et al., 2017). Inhibitors of anti-apoptotic Bcl-2 proteins directly inhibit the underlying mechanism of shuttling Bax between mitochondria and cytosol and therefore increase the diversity of cellular responses (Edlich et al., 2011 and Schellenberg et al., 2013). The approach of the manuscript is certainly interesting and corroborates previous findings.

The authors' lack of insight into the Bax literature produces also statements like the suggestion that active caspase induces Bax accumulation and oligomerization on the mitochondria. This is incorrect. Another example is the use of membrane potential-dependent MitoTracker staining and their surprise upon loss of the staining during mitochondrial apoptosis (a well-characterized phenomenon with well-established alternatives suitable for staining mitochondria in apoptotic cells).

Interpreting the data in terms of Bax regulation would imply directly probing for Bax. Without Bax measurements on the mitochondria the manuscript suffers from over-interpretation. Under the chosen title one would expect to be informed about individual contributions by Bax, Bak and Bok or the contribution of death receptor-induced apoptosis by the receptor ligand TRAIL.

In sum, the statistical approach to cell-to-cell variability on Bax regulation presented in this manuscript supports a characterized phenomenon but lacks the previously provided mechanistic insight. The authors could contribute to the field by applying their method to the question, why cells become stress-resistant. Showing stress resistance caused by different concentrations and the interplay of different proteins on the mitochondria using this approach and confirming this under experimental conditions perhaps together with identifying genetic underpinning would have provided a giant leap forward in understanding apoptosis regulation.

Reviewer #4 (Remarks to the Author):

In this work the authors study the relationship between the cell-to-cell variability (CCV) in mitochondrial density and the fractional response to TRAIL (inducing apoptosis). They found that the cell response to TRAIL is in part dependent on mitochondrial density and that this phenomenon is cell-type dependent. Their findings were quantitatively evaluated using a statistical framework they developed to measure the impact of CCV on the cell response. The framework comprises a parameterization of a cumulative distribution function by fitting it to a Hill function, so the first and second moments of the distributions can be obtained. An expression for relative contributions of a specific component to total signal is then derived; with that they quantified the respective role of

mitochondrial density on TRAIL response variability in Jurkat, MDA-MB-231 and HeLa cells. Using a coarse-grained model of apoptosis, they suggest a general mechanism implicating variable abundance of Bcl-2 family proteins at the mitochondria membrane.

The statistical approach would gain greatly if substantiated with other CCV sources within apoptosis signaling and comparisons/references to current statistical approaches used in this context.

An interesting observation is that mitochondrial density, not mass, is important; this could definitely mark a difference with the previous findings reported in HeLa cells (Márquez-Jurado et al. 2018), if these two metrics were comparatively assessed in the present study.

Major comments:

1. There is a misconception regarding the use of cycloheximide in Spencer's study (Spencer et al. 2009), which unfortunately diminishes the rationale for the present study, especially when compared to the similar findings reported in Nature Communications this year (Márquez-Jurado et al. 2018). Cycloheximide inhibits de novo protein translation, so it prevents the de novo protein expression differences, but locks in the already existing differences in protein content between cells (it does not remove these differences). In addition, it does not inhibit protein degradation and its differences between cells. Therefore, the "temporal fluctuations in protein abundance" (li. 66) are still prevalent in cells treated with cycloheximide. And so, the remaining and minor fraction of unexplained depreciation in time of death correlation between sister cells is likely to be due to differences in key protein levels (see (Gaudet et al. 2012)). A very valuable experiment here would be to use the proposed statistical framework to evaluate and compare the other sources of CCV, including caspase-8 noise, etc...

2. The HeLa cells have been shown sensitive to TRAIL in other studies. The reported IC50 with the same HeLa cells and with the same TRAIL is slightly above 63ng/ml in Marquez-Jurado study (Márquez-Jurado et al. 2018) versus more than 300ng/ml in the present study. A possible difference is the apoptotic assay being used. It would be good to clarify this point in order to strengthen the findings made in MDA-MB-231 and Jurkat cells using the same apoptotic assay.

Minor comments:

1. line 1: the title should be more focused on the actual findings.
2. li 24: "has yet to be determined" should read "is still under investigation": the (Márquez-Jurado et al. 2018) paper is one example.
3. li 30, 31: "we demonstrate that [...] may increase" should read "our study suggests that [...] may increase".
4. li 62-67: should be re-written according to major comment #1
5. li 64 : "cyclohexamide" should read "cycloheximide"
6. li 89: "shared no MitoTracker population". Fig 1B suggests that it does share some fraction...
7. li 294: "measuring [...] regardless of TRAIL dose". The cited study actually measures "as a function of" not "regardless of" TRAIL dose, but shows that it is independent.
8. li 313: "Conversely HeLa cells showed no mitochondrial density dependence". Jacob et al. study (Jacob et al. 2016) strongly suggests that the mitochondrial density is correlated to cell death in HeLa cells, and Marquez-Jurado (Márquez-Jurado et al. 2018) shows that mitochondrial mass is correlated to cell death in HeLa cells. This difference should be discussed.
9. Supplementary notes: the parameters r , γ and μ are fitted to qualitatively reproduce observations from Albeck's study (Albeck et al. 2008), which does not correspond well with the experimental conditions described by the authors in Jurkat and MDA-MB-231 cells.
10. General: The novelty of their approach should be discussed and evaluated with other methods

to parametrize and compare distributions.

References

- Albeck, J.G. et al., 2008. Modeling a snap-action, variable-delay switch controlling extrinsic cell death. *PLoS Biology*, 6(12), pp.2831–2852.
- Gaudet, S. et al., 2012. Exploring the contextual sensitivity of factors that determine cell-to-cell variability in receptor-mediated apoptosis. *PLoS Computational Biology*, 8(4), p.e1002482.
- Jacob, S.F. et al., 2016. An Analysis of the Truncated Bid- and ROS-dependent Spatial Propagation of Mitochondrial Permeabilization Waves during Apoptosis. *Journal of Biological Chemistry*, 291(9), pp.4603–4613.
- Márquez-Jurado, S. et al., 2018. Mitochondrial levels determine variability in cell death by modulating apoptotic gene expression. *Nature communications*, 9(1), p.389.
- Spencer, S.L. et al., 2009. Non-genetic origins of cell-to-cell variability in TRAIL-induced apoptosis. *Nature*, 459(7245), pp.428–432.

All changes from the previous version are indicated in blue in the manuscript. Supplemental information was re-written.

Reviewer 1

The problem of relating molecular and phenotypic data is very important. I congratulate the authors to a great contribution to the field. The method, DEPICTIVE, is novel and appealing, amongst others, due to its simplicity.

We thank the reviewer for his enthusiasm in our work.

There are some studies describing fractional data using mechanistic models, e.g. Heinrich et al., Nat. Cell Biol., 15(10), 2013 and Kallenberger et al., Sci. Signal., 7(316), 2014. I think it would be good to discuss this work. It appears that the existing methods are also rather flexible but more complex and do not possess the same beauty and easy applicability.

We thank the reviewer for indicating these references, we now briefly discussed other approaches for describing fractional data as follows:

“Also, previous methods have been developed to quantify how non-genetic CCV of protein levels influence the fractional responses of cell fate decision in cell populations during mitotic checkpoint signaling [Heinrich2013] and apoptosis [Kallenberger2014], but none takes advantage of the information available in a full dose response curve to perturbations.”

The assumption that the mitochondria abundance is more or less constant seems to be rather important for the application. For this reason, I would suggest to improve Supplement Figure 2. In B and D, that the high population seems to drop below the control and it is unclear how the variability changes. To convince readers that the above assumption is correct, I would suggest to depict the full distributions at the individual time points after sorting. (In additions, the lines shown in D might be removed. The parameterization is unclear and I have a hard time imaging that the best fit is shown.)

Thank you for this important criticism regarding the evidence supporting a foundational assumption in our manuscript. To address this concern we have completely reworked Supplemental Note 2. First, we removed the single cell sorting experiments and instead present the average mitochondria decay dynamics. Second, we have included an entirely new study dedicated to measuring the correlation time scale of mitochondria in single cells. In short, we find the following:

First, we found that mitochondria degradation on average is slow relative to the time scale of our experiments. Specifically, we find that for at a minimum of six hours, the average abundance of mitochondria remains unchanged (our experiment is performed at four hours). Only after 25 hours, 50% of the mitochondria signal has degraded. Here is one panel from Supp. Fig.2

Second, we measured an approximate correlation time-scale of the stochastic process responsible for mitochondria production and degradation. In this experiment we label mitochondria with two spectrally distinct labels, l_1 (Mitotracker Green) and l_2 (Mitotracker Red), at different times. As such, any mitochondria present at time t_0 will be labeled with l_1 , and at time $t_1 = t_0 + \tau$ these original mitochondria as well as any newly generated mitochondria will be labeled with l_2 . In consequence, the correlation between l_1 and l_2 for different values of τ will be informative to the time-scale of the stochastic process responsible for generating and degrading mitochondria. In short, we found that the correlation between l_1 and l_2 marginally decays by $\tau=11$ hours and requires approximately 24 hours to fully decorrelate. Here is one panel from Supp. Fig.3:

We feel that together the average dynamics and the correlation experiments support our assumption that mitochondria abundance does not change significantly over the time-scale of our dose response experiments. We thank the reviewer for his recommendations, and for motivating us to clarify this important point.

The experimental data seem to provide merely information about the CCV in mitochondria density. Accordingly, the possibility of DEPICTIVE to select / rank the most important sources cannot be exploited. It would have been interesting to see this. One simple possibility would be to include forward and side scatter in the experimental data, two parameters which likely to be uninformative.

Indeed, showing our DEPICTIVE analysis on high dimensional single cell data would be a great demonstration of the methodology. We thank the reviewer for his insight into using FSC-A and SSC-A channels for this purpose, as we have already collected these single cell measurements and expect that their corresponding variance explained is low. To demonstrate this we:

1. Generalized the DEPICTIVE description in Supplementary Note 3 for analysis of high dimensional single cell data,
2. Updated our DEPICTIVE python package for easy high dimensional analysis,
3. Re-analyzed all of our experimental results and present mitochondria density and SSC-A density results for each experiment in supplementary figures 5-10
 - a. Note that in the main-text Figure 1 we showed that FSC-A and MitoTracker measurements were correlated. Consequently, we introduced “rho”, the mitochondria density, as our variable of interest. For consistency we applied DEPICTIVE analysis to SSC-A / FSC-A single cell measurement and omitted analysis of FSC-A channel alone.

4. Updated our discussion of the calculation of variance explained in Supplementary Note 3.2.1, and
5. Created Supplementary Tables 1 to 5, that explicitly shows the variance explained by each observable for each experiment for simple comparative analysis.

We thank the reviewer for his careful reading of our manuscript and his insightful and constructive recommendations for its improvement.

In the simulation test shown in Figure 3, x_i seems to be normally distributed. However, in the derivation it was stated that in experiments the distribution is more log-normal. It would be interesting to run the test in this setting. In this context, I was also wondering why the authors did not expand κ on page 6 in the Supplement directly with respect to $\log(x)$.

We now clarified in the Figure 3 legend and Supplementary Note 3.3.4 that the simulated variables represent the natural logarithm of abundances of each biological constituent.

Also, the reviewer raises an interesting question when asking whether we should have expanded κ as expressed on Page 6 of the supplement, and copied below,

$$\kappa = \log(g(\langle x \rangle)) + \sum_{i=1}^N \frac{\partial \log(g(x))}{\partial x} \delta x_i + \dots \quad \text{Eq. 1}$$

in terms of the log as opposed to linear variable. Our rationale of expanding in terms of the linear variables are two fold.

First, we expand Eq. 1 in terms of the linear variables on account that we understand the apoptotic signaling pathway using mass-action chemical kinetics. Formally, these equations are constructed from the products of the linear scale variables, and consequently we assume that the function g is well behaved in terms of the linear scale molecular abundances. Moreover, the application of mass-action chemical kinetics is not without precedent. Huang and Ferrell's seminal (Huang and Ferrell, PNAS 1996) work demonstrated that mass-action kinetic models could predict the response of the MAPK pathway in *Xenopus* extracts, and ever since these models have been successfully applied in studying the apoptotic response [Albeck et al. *PLoS Biology* 2008, Spencer et al. *Nature* 2009], Cancer research [Bouhaddou et al. *PLoS Comp Bio* 2018], etc.

Our second reason for expanding Eq. 1 in terms of the linear variables is out of mathematical transparency. To see this, consider the higher order terms of the expansion in Eq. 1 with respect to the i^{th} molecular species and for simplicity let g' represent the first derivative with respect to the log variable, g'' the second, etc.,

$$\kappa - \log(g(\langle x \rangle)) = x_i^{-1} g' |_{x_i=\langle x_i \rangle} \delta x_i + x_i^{-2} [g'' - g'] |_{x_i=\langle x_i \rangle} \delta x_i^2 + x_i^{-3} [g''' - 3g'' + 2g'] |_{x_i=\langle x_i \rangle} \delta x_i^3 + \dots$$

which we simplify,

$$\kappa - \log(g(\langle x \rangle)) = \langle x \rangle_i^{-1} k_1 \delta x_i + \langle x \rangle_i^{-2} k_2 \delta x_i^2 + \langle x \rangle_i^{-3} k_3 \delta x_i^3 + \dots \quad \text{Eq. 2}$$

Then if we were to expand Eq. 1 in terms of the log,

$$x - \log(g(e^{\langle y \rangle})) = g' \delta y_i + g'' \delta y_i^2 + g''' \delta y_i^3 + \dots$$

Eq. 3

In our manuscript we assume that the fluctuations of molecular components are small with respect to the mean abundances (over a 4 hour time course in a single cell). Under such a condition, it is easy to see from Eq. 2 that truncating the series to first order is appropriate, while more work is required to truncate the series in Eq 3.

In conclusion, we hope that the reviewer appreciates our reasoning behind our formal expansion of the single cell sensitivities.

Supp. p.6: Are the subscript in the equation before Eq. 9 correct or should “m” be “rho”? For the derivation of (17) some additional comments might be valuable.

Thank you for informing us of our error, indeed the subscripts in the equation prior to Eq. 9 were mistakenly set to “m” as opposed to “rho”. We hope that this error did not cause too much confusion in your original assessment of our manuscript.

As noted in our response regarding the use of side-scatter measurements to test our method, we have rewritten these sections to be more general.

1. Now, when discussing the variance of single cell sensitivities attributable to a single component, we no longer refer to “rho” but component “j” and the corresponding random variable modeling its abundance among single cells “x_j”.
2. In the development of the single cell dose response we refer to a set of observed components. The corresponding values to this observed set are the coefficients encoded in a column k_x , the log-scaled variances are now a covariance matrix C, and the Hill coefficient n_x .

We hope that this new formulation not only error free, but more transparent in terms of applying to new experimental measurements.

We also detailed in Supplementary Note 3.3 the derivation of the equations for fitting the Hill model.

Supp. p.7: I think it would be helpful to explain why the IC20 values was chosen.

This threshold was chosen due to the variability in mitochondria staining across samples. With the new approach we now use IC15 instead. We added the following explanation to the supplement:

“We chose the IC15 due to the variability in mitochondria staining across samples. With this approximate joint distribution we are able to apply our fitting protocol.”

Supp. Fig. 3 and later: Labeling seems to be inconsistent with text. “\pi()” is used instead of “p()”.

We thank the reviewer for finding this source of confusion for the reader. As such, we have made the labels in figures consistent with those of the main-text.

The mechanistic model presented in the section “Bax concentration dependence on mitochondria surface area” possesses several parameters and it is not really clear how well the parameters can be estimated from the available data.

We thank the reviewer for identifying this important point, as it is often the case in mechanistic biological modeling that the number of model parameters far exceed that which can be inferred from the available data. Indeed, our mechanistically inspired coarse-grained model is not an exception, and our approach to this challenge is multifaceted. First, we make specific and physically inspired approximations allowing us to group parameters and unknowns into a smaller set of meta-parameters and express variables as dimensionless entities. This strategy allowed us to reduce the original 19 kinetic parameters to 11 meta parameters. In addition, we reduce the dynamic equations of the four distinct proteins - initiator caspase, effector caspase, Bax/Bak and Bcl-2 - and the distinct k -mers of Bax/Bak on the mitochondria surface to one dynamic variable and one algebraic variable. Next, we perform semi-quantitative analysis to match the non-linear dynamic properties perceptible in bifurcation diagrams to the phenomena observed in our single cell measurements. Lastly, is to infer a subset of parameters from pre-existing single-cell temporal measurements of initiator and effector caspase activity.

In terms of parameter inference from data we infer four meta-parameters from the single cell temporal measurements originally cited in Albeck et al. These parameters are “ y_{τ} ” the effective total amount of initiator caspase (IC), “ r ” the effective rate of spontaneous generation of activate initiator caspase, “ μ ” the effective positive feedback local to the receptor and modeled as an autocatalytic reaction, and “ γ ” the effective positive feedback from the effector caspase. First, we assume that the dynamics of the IC signal in the normalized temporal measurements becomes stationary after MOMP on account that all the IC molecules have been activated. In consequence we estimate y_{τ} by taking the average IC measured in each cell at the last time point of the series. We then infer the remaining three parameters by fitting our dynamic model to the temporal measurements.

Indeed, even in this relatively simple case inferred parameters of biologically inspired nonlinear dynamic models can be difficult to estimate and in some cases lead to spurious results. Upon review, we found that the profile likelihood, as documented in Raue et al., to be a good strategy for estimating parameter confidence intervals, and determining whether our reported values represent one of several possible solutions or their uncertainty is so large that our results are meaningless.

We have appended Supplemental Note 6.4 titled “Inferring dynamic model parameters” with our results and display them below for your convenience. The corresponding profile likelihood confirmed that the reported parameters are optimal and unique on the interval 5 times above and below their optimal value. Moreover, from these profile likelihoods we could easily compute a 95% confidence interval for parameters γ and μ , but for r could only establish an upper-bound. While the $\Delta\chi^2$ for values of r below the optimal value of r never reaches the threshold required for 95% confidence, it is bounded by zero as negative parameter values of our model are unphysical. In consequence, our estimation of these parameters seems reasonable for the application.

In the context of reusability I was wondering whether the authors plan to release some software package. This would definitely boost the impact further and would allow other researchers to continue along this profitable route

Yes, we have developed a python package for both the mechanistic mitochondria simulations and our DEPICTIVE strategy for estimating the phenotypic variance explained by biological constituents. We plan to release the software pending acceptance and publication of our manuscript.

References:

Albeck JG, Burke JM, Spencer SL, Lauffenburger DA, Sorger PK (2008) Modeling a Snap-Action, Variable-Delay Switch Controlling Extrinsic Cell Death. PLoS Biol 6(12): e299.

Bouhaddou M, Barrette AM, Stern AD, Koch RJ, DiStefano MS, Riesel EA, et al. (2018) A mechanistic pan-cancer pathway model informed by multi-omics data interprets stochastic cell fate responses to drugs and mitogens. PLoS Comput Biol 14(3): e1005985.

Raue A., C Kreutz, T Maiwald, J Bachmann, M Schilling, U Klingmüller, ... (2009) Structural and practical identifiability analysis of partially observed dynamical models by exploiting the profile likelihood. Bioinformatics 25 (15), 1923-1929

SpencerSL(1), Gaudet S, Albeck JG, Burke JM, Sorger PK. Nature. 2009 Non-genetic origins of cell-to-cell variability in TRAIL-induced apoptosis. May 21;459(7245):428-32

Reviewer 3

Meyer and colleagues study cell-to-cell variability to TRAIL-induced apoptosis in a statistical approach. Seemingly overwhelmed by the large body of Bax literature the authors missed a number of essential publications altogether.

To be sure, we first want to clarify that the main goal of this manuscript was not to exhaustively describe and model the extensive amount of literature regarding apoptosis, but to develop a coarse grained dynamic model to help interpret the sources of cell to cell variability in cell death. We tried to be careful citing adequate articles, indeed with shortcomings. For the particular case of Bax/Bak retro-translocation, we thank the reviewer for indicating the important phenomenon now cited in the manuscript. Given that our simplified mechanistic model assumes that mitochondria have only one membrane and the reactions are assumed to be at equilibrium, we consider that the specific retro-translocation caused by Bcl-xL is accounted for when we consider the equilibrium between cytoplasmic and mitochondrial Bax/Bak states (see also eq. 21 in the supplementary materials).

Previously, variable effective concentrations of the pro-apoptotic protein Bax at the mitochondria have been shown to determine the fractional cell death response (Todt et al., 2013 and Todt et al., 2015) and to correlate with cell survival of leukemia patients (Reichenbach et al., 2017).

Although previous studies suggested that Bax/Bak accumulation at mitochondria leads to an increased predisposition to death by apoptosis [Todt et al], the interpretation of our results is that surviving cells with higher mitochondria density effectively dilute the concentration of mitochondria-associated Bax/Bak, thus affecting the proapoptotic activity of Bax/Bak localized at the mitochondria. We now provide further evidence that our interpretation is correct as surviving cells with higher mitochondria density levels do not have lower Bak levels as might be expected (see below for a full description of experiments in Supp. Fig. 15 and 16).

Inhibitors of anti-apoptotic Bcl-2 proteins directly inhibit the underlying mechanism of shuttling Bax between mitochondria and cytosol and therefore increase the diversity of cellular responses (Edlich et al., 2011 and Schellenberg et al., 2013). The approach of the manuscript is certainly interesting and corroborates previous findings.

We are glad that this Reviewer finds our results interesting, and we think that our observations regarding the combination of TRAIL and ABT263 not only corroborate previous findings, but also quantitatively explain the unmasking of the mitochondrial-density dependence in MDA-MB231 cells survival, already present in Jurkat cells and altogether absent in HeLa cells. This shows that such combination can have different effects in different cell lines (see Figure 5). Our interpretation for this effect is in line with what this referee suggests in that different combinations of 'BH3-only' proteins lead to differences in the effects of their interactions with Bax/Bak that can lead to changes in their localization and the live/dead outcome in different cell lines. You can see this implemented in the fact that the main difference between the Jurkat and MDA-MB231 cell mathematical models is a decrease in the binding affinity between Bcl2 and Bax/Bak (see Supp. Table 6). In support of these results, we have added new supplemental data to the manuscript. Supplemental Figure 14 (see below) shows Western immunoblots of whole cell lysates showing that Bax and Bak are predominantly expressed in MDA-MB231 cells, while only Bak is present in Jurkat (also previously observed in [Brimmel 1998]). Our data further suggests that Bcl-2 is mostly expressed in Jurkat cells, while Bcl-xL is the main anti-apoptotic Bcl-2-family member in MDA-MB231 cells.

Here is Supp. Fig.14:

The authors' lack of insight into the Bax literature produces also statements like the suggestion that active caspase induces Bax accumulation and oligomerization on the mitochondria. This is incorrect.

We again apologize for taking too many shortcuts when describing apoptosis and appreciate this knowledgeable comment from the Reviewer and modified the text to reflect a more detailed mechanistic view of TRAIL-induced apoptosis, in line with established literature. The revised section of the manuscript is pasted below for convenience:

“During extrinsic apoptosis, TRAIL stimulates cell death by binding to its cognate death receptors on the cell surface forming a complex that activates Caspase 8 (Figure 1A), the so-called initiator caspase (IC). Active IC activates pro-apoptotic BH3-only proteins, which, directly or indirectly, activate pro-apoptotic Bcl-2 family proteins Bax/Bak. Active Bax/Bak can commit a cell to apoptosis by translocating from the cytosol to the outer mitochondrial membrane where they oligomerize and form pores [22, 23], which allow for the diffusion of pro-apoptotic molecules from the intermembrane space of the mitochondria into the cytosol [24, 25]. The pro-apoptotic activities of Bax/Bak are counteracted by pro-survival Bcl-2 proteins such as Bcl-xL, which constantly retro-translocates Bax/Bak from the mitochondria back into the cytosol [Edlich et al., 2011 and Schellenberg et al., 2013], thus protecting cells from committing to apoptosis by shifting relative subcellular localization of Bax/Bak [Todt et al 2013, Todt et al 2015]. “

Another example is the use of membrane potential-dependent MitoTracker staining and their surprise upon loss of the staining during mitochondrial apoptosis (a well-characterized phenomenon with well-established alternatives suitable for staining mitochondria in apoptotic cells).

It is unfortunate that the Reviewer interprets as surprise our decision to explicitly state that MitoTracker deep red is a voltage dependent dye. We did it so that readers not familiarized with

the field, and given the wide audience of this publication, may understand the rationale for our analysis, heavily dependent on this indeed well-documented fact. We are also familiar with alternatives (see usage of voltage-independent MitoTracker green in Supp. Note 2), but given the importance of obtaining a robust measure of mitochondria density, we decided not to use the voltage-independent MitoTracker dyes.

Interpreting the data in terms of Bax regulation would imply directly probing for Bax. Without Bax measurements on the mitochondria the manuscript suffers from over-interpretation.

We agree with the Reviewer that probing for Bax/Bak in cells exposed to different concentrations of TRAIL would strengthen our interpretation of the mitochondrial effect we observed. To address the Reviewer's point, we performed additional experiments that now provide in both cell lines concomitant measurements of mitochondria levels by FACS and relative Bak levels in the remaining fraction of surviving cells measured by Western blot (see Supp. Fig. 15 and 16). We chose to measure Bak which is expressed in both MDA-MDB231 and Jurkat cells (see Supp. Fig. 14) and also because Bak localization correlates with mitochondria-associated Bax [Reichenbach et al 2017, Korsmeyer et al 2000].

Briefly, cells were stained with MitoTracker, followed by treatment with 7 doses of TRAIL for 4h. Then, cells were stained with fluorescently labeled Annexin V, and the population of live cells were sorted by FACS and collected to generate whole cell lysates for Western blot. MitoTracker levels measured during FACS collection were used to compute the probability distribution of mitochondria density for each dose of TRAIL, and Western blots were probed for Bak levels and beta Actin, which was used as control to normalize Bak signal across all samples [Supp. Fig. 15]. In order to make a meaningful comparison of the relative contributions of mitochondria and Bak levels to cell death, we computed the averages of mitochondria density in surviving cells measured by FACS and plotted them alongside with the average signal of normalized Bak/Actin levels measured by Western blot, for each fraction of cells alive caused by increasing doses of TRAIL (see Supp. Fig. 15 and 16).

Here 2 panels from Supp. Fig. 15 for Jurkat cells:

In line with our results, cells that survived exposure to increasing doses of TRAIL exhibited increasingly higher mitochondria density (see Supp. Fig. 15 and 16). If as we argue, mitochondrial density variability is a significant driver of stochastic cell survival, the average signal of normalized Bak would be depicted as a flat line across all TRAIL doses. In contrast, if Bak was the main factor driving fractional killing, then we would expect surviving cells to show decreased levels of Bak. We found that relative Bak levels were flat for both Jurkat and MDA-MDB-231 cells (see Supp. Fig. 15 and 16) except for a highly variable increase at higher doses of TRAIL for Jurkat cells. In summary, we think that mitochondria density being the main driver

for cell survival not only corroborates but is complementary to the results by Todt *et al* where Bax/Bak mitochondrial localization is the main driver for cell death. Here are 2 panels from Supp. Fig.16 for MDA-MDB-231 cells:

Under the chosen title one would expect to be informed about individual contributions by Bax, Bak and Bok or the contribution of death receptor-induced apoptosis by the receptor ligand TRAIL.

We thank the reviewer for this comment and agree that controlling for the role of pro-apoptotic Bcl-2 proteins is necessary as our method computes the mitochondria-explained variance by averaging over each other contributing factor, hence our measurements represent an upper bound, given that correlations in proteins and mitochondria density will diminish the unique influence of mitochondria (see next reply for further discussion on this point). We already described on the previous paragraph measurements regarding the influence of Bax/Bak. Furthermore, to our knowledge, the existing Bok literature suggests that expression of Bok is tissue-specific, mostly limited to testis, ovary, and uterus [Hsu *et al* 1997]. In contrast, expression of Bax and Bak is known to be widespread in mammalian tissues [Wei *et al* 2001]. Bok can directly cause MOMP upon dysregulation of ER-associated degradation pathway components, and Bok stability is regulated by the ERAD pathway rather than BCL-2 family members [Llambi *et al* 2016]. Previous studies suggest that the role of Bok in cell death is mostly limited to ER-related stress, where it is targeted for degradation by the proteasome. Thus, studying the contribution of Bok to cell death by apoptosis is beyond the scope of this manuscript, as is the death receptor whose contribution has already been studied by Gaudet *et al*. Because we believe to have proven that mitochondrial density is a large contributor to cell to cell variability in cell survival we thought the current title truly reflected that. However, to address the Reviewer's concern, we could modify the title to "Mitochondrial origins of fractional control in regulated cell death".

In sum, the statistical approach to cell-to-cell variability on Bax regulation presented in this manuscript supports a characterized phenomenon but lacks the previously provided mechanistic insight.

We thank the Reviewer for this insightful comment and provide further clarification on our perspective. We understand the importance of the detailed mechanisms highlighted by the Reviewer about the sequence of all the individual steps taking place during apoptosis. Previous studies have captured the high variability of the outcome of TRAIL treatment by measuring the concentrations of multiple proteins in the apoptosis pathway and modeling each individual step and interplay between them [Albeck *et al* 2008, Marquez-Jurado *et al* 2018]. However, these

approaches failed to identify one single factor driving the divergence in cell fate and attributed the variability in cell death to cell-to-cell differences in the levels of multiple individual proteins in the pathway.

To overcome these challenges, we developed a new method, namely DEPICTIVE, that bypasses the need for measurements of multiple individual proteins in the apoptosis pathway machinery, and still pinpoints some of the high variability in fractional killing associated with a single source, the mitochondrial density. In our model we assessed the initial state of a cell in the live/death phase plane based upon one single parameter, the mitochondria density, and predicted if and when that cell transitioned to another state (from live to dead) in response to TRAIL. Thus, we believe that our innovative systematic and quantitative approach is complementary to existing models, provides a conceptual advance to the understanding of CCV in regulated cell death through DEPICTIVE, and can be applied to any other discrete system in the absence of mechanistic insights.

The authors could contribute to the field by applying their method to the question, why cells become stress-resistant. Showing stress resistance caused by different concentrations and the interplay of different proteins on the mitochondria using this approach and confirming this under experimental conditions perhaps together with identifying genetic underpinning would have provided a giant leap forward in understanding apoptosis regulation.

We agree that the Reviewer brings up a highly interesting question hovering the cell death field, and we believe that our method could potentially contribute significantly to identifying and ranking the sources of variability at play when cells become stress-resistant. Studies by Flusberg *et al* and others have tackled this question and even attempted to provide an interpretation based on gene expression data that suggested that sustained activation of innate immunity factors might correlate with emergence of drug resistance. However, we believe further exploration of that question is beyond the scope of our current manuscript.

References:

Albeck JG, Burke JM, Spencer SL, Lauffenburger DA, Sorger PK (2008) Modeling a Snap-Action, Variable-Delay Switch Controlling Extrinsic Cell Death. *PLoS Biol* 6(12): e299.

Brimmell M, Mendiola R, Mangion J, and Packham G.. BAX frameshift mutations in cell lines derived from human haemopoietic malignancies are associated with resistance to apoptosis and microsatellite instability. *Oncogene*. 1998 Apr 9;16(14):1803-12

Chen, Douglas R. Green, (2016) BOK Is a Non-Canonical BCL-2 Family Effector of Apoptosis Regulated by ER-Associated Degradation Cell. 165(2): 421–433.

Flusberg DA, Jérémie Roux, Sabrina L. Spencer, and Peter K. Sorger, Cells surviving fractional killing by TRAIL exhibit transient but sustainable resistance and inflammatory phenotypes. (2013). *Molecular Biology of the Cell*.(24)-2186

Gaudet S, Spencer SL, Chen WW, Sorger PK (2012) Exploring the Contextual Sensitivity of Factors that Determine Cell-to-Cell Variability in Receptor-Mediated Apoptosis. *PLoS Comput Biol* 8(4): e1002482.

Hsu S, Kaipa A, McGee E, Lomeli M and Hsueh A (1997) Bok is a pro-apoptotic Bcl-2 protein with restricted expression in reproductive tissues and heterodimerizes with selective anti-apoptotic Bcl-2 family members. Proc. Natl. Acad. Sci. USA 94: 12401–12406.

Korsmeyer SJ, Wei MC, Saito M, Weiler S, Oh KJ, Schlesinger PH.(2000) Pro-apoptotic cascade activates BID, which oligomerizes BAK or BAX into pores that result in the release of cytochrome c. Cell Death Differ. (12):1166-73.

Llambi F, Yue-Ming Wang, Bernadette Victor, Mao Yang, Desiree M. Schneider, Sébastien Gingras, Melissa J. Parsons, Janet H. Zheng, Scott A. Brown, Stéphane Pelletier, Tudor Moldoveanu, Taosheng

Reichenbach, Wiedenmann C., Schalk E., Becker D., Funk K., Scholz-Kreisel P., Todt F., Wolleschak D., Dohner K., Marquardt JU., Heidel F. and Edlich F. (2017) Mitochondrial BAX determines the predisposition to apoptosis in human AML. CCR-16-1941

Todt, F, Cakir, Z, Reichenbach, F, Youle, R. & Edlich, F. The C-terminal helix of Bcl-x L mediates Bax retrotranslocation from the mitochondria. Cell death and differentiation 20, 333 (2013).

Todt, F. et al. Differential retrotranslocation of mitochondrial Bax and Bak. The EMBO journal 34, 67–80 (2015).

Wei MC, Zong WX, Cheng EH, Lindsten T, Panoutsakopoulou V, Ross AJ, Roth KA, MacGregor GR, Thompson CB and Korsmeyer SJ (2001) Proapoptotic BAX and BAK: a requisite gateway to mitochondrial dysfunction and death. Science 292: 727–730.

Reviewer 4

The statistical approach would gain greatly if substantiated with other CCV sources within apoptosis signaling and comparisons/references to current statistical approaches used in this context.

We thank the Reviewer for this suggestion. Previous studies have captured the high variability of the outcome of TRAIL treatment by measuring the concentrations of multiple proteins in the apoptosis pathway and modeling multiple individual steps leading to apoptosis [Albeck et al 2008, Marquez-Jurado et al 2018]. However, those approaches failed to identify one single factor driving cell fate and attributed the variability in cell death to cell-to-cell differences in the levels of multiple individual proteins in the pathway. In this manuscript we have found that mitochondria density is a single factor that explains by itself more than 30% of cell-to-cell variability to TRAIL-induced apoptosis. Probing for the other sources of CCV that this reviewer suggests is equivalent to performing a negative control and thus we have added new supporting evidence to our manuscript showing that when applying DEPICTIVE to experimental measurements of FACS Side Scatter values (SSC), the variance explained is as expected minimal (see Supplementary Table 1 and also view response to Reviewer 1).

An interesting observation is that mitochondrial density, not mass, is important; this could definitely mark a difference with the previous findings reported in HeLa cells (Marquez-Jurado et al. 2018), if these two metrics were comparatively assessed in the present study.

As insightfully observed by the Reviewer, Marquez-Jurado *et al* interpret fractional killing as a function of mitotracker mass, we initially performed our analysis using mitochondrial mass but given that mass and cell size are correlated ~ 0.5 (see Fig.1D & G), we preferred to exclude effects due to cell size and used mitochondrial density instead. Still, given the relatively high correlation between cell size and mitochondrial mass, the effect we observe in Jurkat and MDA-MB-231 cells would not be reversed to match Marquez-Jurado *et al* observations in HeLa cells if these were interchanged. Indeed, our interpretation is that mitochondrial-associated Bax/Bak molecules also have a lower likelihood to form pores on a larger mitochondria surface area which we think is proportional to mitochondrial density, not mass, exchanging mass and density does not invert its tendency. However, it is possible that Marquez-Jurado *et al* observations could be highly modified when considering mitochondrial density and not mass given that it is known that gene expression levels can change with cell size (see below for a full discussion about the Marquez-Jurado *et al* article).

Major comments

There is a misconception regarding the use of cycloheximide in Spencer's study (Spencer et al. 2009), which unfortunately diminishes the rationale for the present study, especially when compared to the similar findings reported in Nature Communications this year (Márquez-Jurado et al. 2018). Cycloheximide inhibits de novo protein translation, so it prevents the de novo protein expression differences, but locks in the already existing differences in protein content between cells (it does not remove these differences). In addition, it does not inhibit protein degradation and its differences between cells. Therefore, the "temporal fluctuations in protein abundance" (li. 66) are still prevalent in cells treated with cycloheximide. And so, the remaining and minor fraction of unexplained depreciation in time of death correlation between sister cells is likely to be due to differences in key protein levels (see (Gaudet et al. 2012)). A very valuable experiment here would be to use the proposed statistical framework to evaluate and compare the other sources of CCV, including caspase-8 noise, etc...

We don't think our interpretation of the decay in correlation between sister cells described by Spencer *et al.* is a crucial point of our manuscript as we use it only as a minor argument when discussing the results of Marquez-Jurado *et al.* Indeed, we came to the conclusion that Marquez-Jurado *et al* are measuring a different phenomenon, namely mitochondrial content in HeLa cells that died. If this reviewer still thinks this is an important point we are willing to edit out this paragraph.

The HeLa cells have been shown sensitive to TRAIL in other studies. The reported IC50 with the same HeLa cells and with the same TRAIL is slightly above 63ng/ml in Marquez-Jurado study (Márquez-Jurado et al. 2018) versus more than 300ng/ml in the present study. A possible difference is the apoptotic assay being used. It would be good to clarify this point in order to strengthen the findings made in MDA-MB-231 and Jurkat cells using the same apoptotic assay.

We thank the Reviewer for this comment and insightful suggestion and now we provide further clarification on our perspective (also please refer to Supp. Note 5). The flow cytometry apoptotic assay we use in our study is widely established in the cell death field and is featured in a number of publications as a standard assay for determining the IC50 of various death-inducing agents [Rieger et al 2011, Crowley et al 2015]. However, we believe there are several other differences between our apoptotic assays that can justify the differences in the observed sensitivity of cells to TRAIL.

First, the TRAIL used in the Marquez-Jurado et al and the Spencer et al studies is not the same that we used in our study. As properly referenced in our Materials and Methods, we used Superkiller TRAIL, which has been shown to significantly improve trimerization and activation of

death receptors TRAIL-R1 and -R2 on the cell surface and triggering the assembly of the death-inducing signaling complex (DISC) by effectively mimicking the clustering of those receptors upon binding of natural death-inducing stimuli [Johnstone et al 2008, Mandal et al 2014, Naimia et al 2018]. In those studies, TRAIL was used in combination with cycloheximide to suppress the activation of parallel translation-dependent survival signals via non-apoptosis-inducing receptors [Falschlehner et al 2007, Hellwig et al 2008], which as shown in Spencer et al Supplementary Figure 4 increases sensitivity to TRAIL, consistent with previous reports [Wajant et al 2000, Wang et al 2008]

Second, it is widely known that the IC₅₀ of TRAIL is a function of time [Thomas and Hersey 1998, Zhang et al 2005, Trivedi and Mishra 2015]. In the Marquez-Jurado assay, cells were treated with TRAIL for 24h, while in our study we treated cells for 4h, which is under the timescale for natural mitochondria generation and turnover dynamics to interfere with the effective abundance of mitochondria in a cell during the time course of the experiment. As shown in our *Supplementary Note 2*, mitochondria density does not change significantly within 4h, but at 24h it is different than at time zero, likely due to cell division, mitogenesis, mitochondria turnover, and remodeling of mitochondria network through fission and fusion events. Notably, none of the above-mentioned studies has taken mitochondria dynamics into account, which we believe might be at the source of some of the discrepancies.

Third, in contrast to the other models that extracted IC₅₀ values from the absolute coordinates of fractional response to TRAIL, in our model we computed the effective IC₅₀ of TRAIL by using the maximum amplitude of the cell death response to TRAIL, which we believe reflects the effective outcome of TRAIL signaling. The difference in IC₅₀ values between our studies is simply a consequence of our model taking into account the TRAIL signal as a function of the final outcome, in a way that captures the entire dynamics of the cell response as a whole. Indeed, when we calculate TRAIL IC₅₀ for HeLa cells as done by Marquez-Jurado, we obtain an IC₅₀ ~80ng/ml.

Fourth, as pointed out by the Reviewer, HeLa cells are in fact sensitive to TRAIL. However, as shown in Figure 1 of Marquez-Jurado as well in our Supplementary Figure 11, treatment of HeLa cells with maximum dose of TRAIL incurs in a maximum cell death fraction of approximately 50% in all studies alike. Perhaps this is indicative of the contribution of other biological mechanisms of death alternative to extrinsically regulated apoptosis. Previous studies have shown that HeLa cells do not express Parkin [Denison et al 2003, Pawlyk et al 2003] and by lacking mitophagy machinery their response to death-inducing ligands will be affected, an interesting phenomenon that is beyond the scope of this study.

Finally, and most importantly, Marquez-Jurado et al. are able to use single cell tracking to link a cell's mitochondria abundance at time 0 with its fate at 24 hours. With this experimental design they were able measure the mitochondria abundances for cells that remained alive and those that transitioned to the dead state. In Figure 2 of their manuscript they show that the average mitochondria abundance for cells that transition from live to dead decreases with TRAIL. Moreover, by qualitative inspection the mean mitochondria abundance among the live cells remains relatively constant, just as we measured in HeLa cells (see Sup.Fig.11 and Sup Note 5). Indeed, as TRAIL increases, the average mitochondria abundance of live and dead cells converges. While a truly interesting insight, our experimental design fundamentally cannot measure the phenomena they report. If, we were able to measure the mitochondria abundance of the cells that transitioned from live to dead states, perhaps we would have come to the identical conclusion for the HeLa cell experiment. In consequence, we believe that our results and those of Marquez-Jurado et al. complement one another.

Minor Comments

line 1: the title should be more focused on the actual findings.

We changed the title to the more specific “Mitochondrial origins of fractional control in regulated cell death”.

li 24: “has yet to be determined” should read “is still under investigation”: the (Márquez-Jurado et al. 2018) paper is one example.

We changed as suggested by this reviewer.

li 30, 31: “we demonstrate that [...] may increase” should read “our study suggests that [...] may increase”.

We changed as suggested by this reviewer.

- li 62-67: should be re-written according to major comment #1

li 64 : “cyclohexamide” should read “cycloheximide”

We changed as suggested by this reviewer.

li 89: “shared no MitoTracker population”. Fig 1B suggests that it does share some fraction...

We changed to “shared almost no MitoTracker population”

li 294: “measuring [...] regardless of TRAIL dose”. The cited study actually measures “as a function of” not “regardless of” TRAIL dose, but shows that it is independent.

We changed as suggested by this reviewer to “as a function of of TRAIL dose, but show that the effect is independent of the dose”

li 313: “Conversely HeLa cells showed no mitochondrial density dependence”. Jacob et al. study (Jacob et al. 2016) strongly suggests that the mitochondrial density is correlated to cell death in HeLa cells, and Marquez-Jurado (Márquez-Jurado et al. 2018) shows that mitochondrial mass is correlated to cell death in HeLa cells. This difference should be discussed.

We discussed the differences between Marquez-Jurado *et al* on a major comment above, briefly as shown in Figure 2A of their manuscript, the mitochondrial dependency they observe across TRAIL doses is for death cells as in live cells mitochondria levels seem to be constant as observed in our experiments (see Sup. Note 5 and Sup.Fig 11). Regarding Jacob *et al*, their study does not measure mitochondria density, but models its effects on the fast propagation of internal waves of MOMP. However, in their model, mitochondria density has opposite effects on tBid and ROS waves and its overall effect/dependencies on cell death is far from clear.

Supplementary notes: the parameters r , γ and μ are fitted to qualitatively reproduce observations from Albeck’s study (Albeck et al. 2008), which does not correspond well with the experimental conditions described by the authors in Jurkat and MDA-MB-231 cells.

As there are no measurements for the dynamics of Initiator and effector caspases in the cell lines we used, we assumed that the general shape of the dynamics obtained through quantitative fits are the best proxy available.

General: The novelty of their approach should be discussed and evaluated with other methods to parametrize and compare distributions.

The detailed application of our DEPICTIVE approach to other data can be another paper in itself and consequently is outside the scope of this manuscript.

References

Crowley LC, Marfell BJ, Scott AP, Waterhouse NJ. 2015. Quantitation of apoptosis and necrosis by annexin V binding, propidium iodide uptake, and flow cytometry. *Cold Spring Harb Protoc*.

Denison, S.R., F. Wang, N.A. Becker, B. Schule, N. Kock, L.A. Phillips, C. Klein, and D.I. Smith. 2003. Alterations in the common fragile site gene Parkin in ovarian and other cancers. *Oncogene*. 22:8370–8378.

Falschlehner C, Christoph H. Emmerich, Björn Gerlach, and Henning Walczak, TRAIL signalling: Decisions between life and death. 2007. TRAIL signalling: Decisions between life and death. *The International Journal of Biochemistry & Cell Biology Volume 39 (7) 1462-1475*

Hellwig CT, Barbara F. Kohler, Anna-Kaisa Lehtivarjo, Heiko Dussmann, Michael J. Courtney, Jochen H. M. Prehn and Markus Rehm. 2008. Real Time Analysis of Tumor Necrosis Factor-related Apoptosis-inducing Ligand/Cycloheximide-induced Caspase Activities during Apoptosis Initiation. *Journal of Biological Chemistry*. 283, 21676-21685.

Johnstone, R. W. et al. The TRAIL apoptotic pathway in cancer onset, progression and therapy. *Nature Reviews Cancer* 8, 782-798 (2008)

Mandal (2014). pERK 1/2 inhibit Caspase-8 induced apoptosis in cancer cells by phosphorylating it in a cell cycle specific manner. *Mol. Oncol.* ,8, 232.

Márquez-Jurado, S. et al. Mitochondrial levels determine variability in cell death by modulating apoptotic gene expression. *Nature communications* 9, 389 (2018).

Naimia A, Ali Akbar Movassaghpourc, Majid Farshdousti Haghc, Mehdi Talebic, Atefeh Entezarid, Farhad Jadidi-Niaraghe, Saeed Solalic. TNF-related apoptosis-inducing ligand (TRAIL) as the potential therapeutic target in hematological malignancies. *Biomedicine & Pharmacotherapy* 98 (2018) 566–576.

Pawlyk, A.C., B.I. Giasson, D.M. Sampathu, F.A. Perez, K.L. Lim, V.L. Dawson, T.M. Dawson, R.D. Palmiter, J.Q. Trojanowski, and V.M. Lee. 2003. Novel monoclonal antibodies demonstrate biochemical variation of brain parkin with age. *J. Biol. Chem.* 278:48120–48128.

Rieger, A. M., Nelson, K. L., Konowalchuk, J. D., & Barreda, D. R. (2011). Modified annexin V/propidium iodide apoptosis assay for accurate assessment of cell death. *Journal of visualized experiments: JoVE*, , (50).

Thomas, W. D & Hersey, P. (1998) TNF-Related Apoptosis-Inducing Ligand (TRAIL) Induces Apoptosis in Fas Ligand-Resistant Melanoma Cells and Mediates CD4 T Cell Killing of Target Cells. *The Journal of Immunology* vol. 161 no. 5 2195-2200

Trivedi R, and D.P. Mishra, Trailing TRAIL resistance: novel targets for TRAIL sensitization in cancer cells, *Front. Oncol.* 5 (2015) 69.

Wang L, Fenghe Du, and Xiaodong Wang. 2008. TNF- α Induces Two Distinct Caspase-8 Activation Pathways. *Cell.* 133, 693–703

Wajant H, Elvira Haas, Ralph Schwenzer, Frank Muhlenbeck, Sebastian Kreuz, Gisela Schubert, Matthias Grell, Craig Smith, and Peter Scheurich. 2000. Inhibition of Death Receptor-mediated Gene Induction by a Cycloheximide-sensitive Factor Occurs at the Level of or Upstream of Fas-associated Death Domain Protein (FADD). *Journal of Biological Chemistry.* Vol 275 (32). 24357–24366.

Zhang, L. & Fang. Mechanisms of resistance to TRAIL-induced apoptosis in cancer. *Cancer Gene Therapy* 12, 228–237 (2005)

REVIEWERS' COMMENTS:

Reviewer #1 (Remarks to the Author):

In the revised version of the manuscript, the authors addressed all my comments. I appreciate the additional work, in particular the application of DEPICTIVE to higher dimensional data (mitochondria density and SSC-A / FSC-A). The results of this are interesting and indicate the relevance of mitochondria density. However, the results also show that in certain cases (e.g. Table 4 and 5), SSC-A / FSC-A can explain a higher portion of the variance than the mitochondria density. Therefore, I suggest as a minor revision that the authors discuss this.

In addition, I would like to point out that the profile likelihoods show that the parameter r cannot be inferred reliably. It is practically non-identifiable as the statistical threshold is not exceeded for values below the optimum. Accordingly, the confidence interval for r is unbounded below. While this should not be critical for the authors' results, I ask them to clarify this. Similarly, I would ask the authors to remove that statement that profile likelihoods address the problem of irregular likelihoods, as profile likelihoods are actually an analysis tool.

Reviewer #3 (Remarks to the Author):

The authors have addressed my concerns with a substantial effort to strengthen their case. Although several issues with presented data and provided discussion remain unresolved, I believe none of these issues undermines their finding as a whole. Therefore, I recommend accepting this manuscript for publication in the interest of a concise reviewing process.

Reviewer #4 (Remarks to the Author):

Overall the study gains in clarity when read with the actual responses provided in the rebuttal. Therefore, it is necessary to include all the newly discussed points in the manuscript, as well as the proposed edits that have not been made yet, namely:

1. Discuss the first two general comments ("other sources of CCV" and "density vs. mass") in the discussion.
2. Edit out paragraph as proposed in response to Major comment 1
3. Add the explanation for IC50 differences of Major Comment 2 (perhaps in Supplemental materials)
4. The response to the "minor point on li 313" and the "Supplementary notes", as they both also adds in clarity.

Reviewer 1

In the revised version of the manuscript, the authors addressed all my comments. I appreciate the additional work, in particular the application of DEPICTIVE to higher dimensional data (mitochondria density and SSC-A / FSC-A). The results of this are interesting and indicate the relevance of mitochondria density. However, the results also show that in certain cases (e.g. Table 4 and 5), SSC-A / FSC-A can explain a higher portion of the variance than the mitochondria density. Therefore, I suggest as a minor revision that the authors discuss this.

We added in the discussion the following sentence:

Remarkably, Bcl-2 inhibition alone increased the variance of sensitivities attributable to mitochondria density from ~0% to between 10% and 25% (Figure 6F), although for the lower value the variance attributed to the internal SSC control was barely lower (see Supplementary Note 4.2 Supplementary Tables 2,3 and 4).

In addition, I would like to point out that the profile likelihoods show that the parameter r cannot be inferred reliably. It is practically non-identifiable as the statistical threshold is not exceeded for values below the optimum. Accordingly, the confidence interval for r is unbounded below. While this should not be critical for the the authors' results, I ask them to clarify this.

We added the following explanation:

Although the profile likelihood for r is practically non-identifiable as very shallow and unbounded below, given that for the biological interpretation of the model we were only interested in positive values, the origin was considered as a lower bound.

Similarly, I would ask the authors to remove that statement that profile likelihoods address the problem of irregular likelihoods, as profile likelihoods are actually an analysis tool.

We replaced:

“One solution to this problem”

with

One way to analyze the extent of the log-likelihood function irregularities

Reviewer 3

The authors have addressed my concerns with a substantial effort to strengthen their case. Although several issues with presented data and provided discussion remain unresolved, I believe none of these issues undermines their finding as a whole. Therefore, I recommend

accepting this manuscript for publication in the interest of a concise reviewing process.

No further changes are requested by this reviewer

Reviewer 4

Overall the study gains in clarity when read with the actual responses provided in the rebuttal. Therefore, it is necessary to include all the newly discussed points in the manuscript, as well as the proposed edits that have not been made yet, namely:

1. Discuss the first two general comments (“other sources of CCV” and “density vs. mass”) in the discussion.

As suggested by this reviewer we added the following in the results section:

while less than ~2% is attributed to side scatter (SSC) that functions as an internal control.

And to the discussion:

In particular, it is possible that Marquez-Jurado observations could be highly modified when considering mitochondrial density and not mass given that it is known that gene expression levels can change with cell size.

2. Edit out paragraph as proposed in response to Major comment 1

The paragraph has now been edited out as suggested by this reviewer.

3. Add the explanation for IC50 differences of Major Comment 2 (perhaps in Supplemental materials)

We added the suggested explanation to the supplementary methods section related to HeLa cells.

The apparent 5-fold higher IC50 we report has to do with the much shorter 4 hour time-span of our experiment and that we computed the effective TRAIL IC50 by using the maximum amplitude of the cell death response to TRAIL, which we believe reflects the effective outcome of TRAIL signaling. Indeed, when we calculate relative TRAIL IC50 for HeLa cells as done by Marquez- Jurado et al., we obtain a much closer IC50 value around 80ng/ml.

4. The response to the “minor point on li 313” and the “Supplementary notes”, as they both also adds in clarity.

Minor point on li 313 was included in the discussion as stated in 1.

As suggested by this reviewer we added these points to the supplementary methods section.

To constrain cell type specific parameter sets for our dynamic model of apoptosis we corroborated the predicted dynamics with previously published data as the best proxy available given there are no measurements for the dynamics of Initiator and effector caspases in the cell lines we used. Specifically, we fit the parameters r , γ , and μ to temporal single cell measurements of a reporter of initiator caspase activity (IC-RP) published in Albeck et al. 2008.